# Structure and replication of *Pseudomonas aeruginosa* phage JBD30

Lucie Valentová (ID), Tibor Füzik (ID), Jiří Nováček (ID), Zuzana Hlavenková (ID), Jakub Pospíšil (ID) & Pavel Plevka (ID) ✉

## Abstract

**Bacteriophages are the most abundant biological entities on Earth, but our understanding of many aspects of their lifecycles is still incomplete. Here, we have structurally analysed the infection cycle of the siphophage *Casadabanvirus* JBD30. Using its baseplate, JBD30 attaches to *Pseudomonas aeruginosa via* the bacterial type IV pilus, whose subsequent retraction brings the phage to the bacterial cell surface. Cryo-electron microscopy structures of the baseplate-pilus complex show that the tripod of baseplate receptor-binding proteins attaches to the outer bacterial membrane. The tripod and baseplate then open to release three copies of the tape-measure protein, an event that is followed by DNA ejection. JBD30 major capsid proteins assemble into procapsids, which expand by 7% in diameter upon filling with phage dsDNA. The DNA-filled heads are finally joined with 180-nm-long tails, which bend easily because flexible loops mediate contacts between the successive discs of major tail proteins. It is likely that the structural features and replication mechanisms described here are conserved among siphophages that utilize the type IV pili for initial cell attachment.**

**Keywords** *Pseudomonas aeruginosa*; Phage; Pili; Structure; Cryo-EM
**Subject Categories** Microbiology, Virology & Host Pathogen Interaction; Structural Biology

## Introduction

Siphophages are the most abundant group of tailed phages that have been identified by sequencing screens (Dion et al, 2020). Despite the high occurrence and variability of siphophages, only a few of them have had their virion structure and replication strategies described (Bebeacua et al, 2013; Sciara et al, 2010; Spinelli et al, 2020; Kizziah et al, 2020; Arnaud et al, 2017; Linares et al, 2023; Zinke et al, 2020; Plisson et al, 2007). Virions of siphophages feature a long flexible non-contractile tail which is connected via a set of head completion proteins and a dodecameric portal to an icosahedral or prolate capsid filled with dsDNA. Phage baseplates

differ depending on the gram index of their hosts and their target receptors, either proteins or sugar moieties (Goulet et al, 2020). The architecture of baseplates of phages that use polysaccharides as receptors, e.g. *Lactococcus lactis* phage TP901-15 (Bebeacua et al, 2013), *Lactococcus lactis* phage P2 (Sciara et al, 2010; Spinelli et al, 2020) and *Staphylococcus aureus* phage 80α (Kizziah et al, 2020), are complex. Their baseplates are decorated with multiple copies of receptor binding proteins, upper baseplate proteins and ancillary proteins. Baseplates of phages utilizing proteinaceous receptors, e.g. *Escherichia coli* phage T5 (Arnaud et al, 2017; Linares et al, 2023) and *Bacillus subtilis* phage SPP1 (Zinke et al, 2020; Plisson et al, 2007), consist of an elongated central fibre with a receptor binding protein at the tip. Bacteriophage T5 has additional L-shaped fibres stemming from the upper baseplate collar, which increase its adsorption efficiency (Linares et al, 2023).

Bacteriophage *Casadabanvirus* JBD30, from the order *Caudoviricetes*, is a temperate phage that infects the bacterium *Pseudomonas aeruginosa* (Bondy-Denomy et al, 2016). Infection by JBD30 relies on binding to type IV pili. JBD30 encodes transposase *gp6* (Appendix Table S1) that enables the random insertion of the phage genome into the host chromosome. Being a prophage, JBD30 protects the lysogenized bacterial cell from superinfection by other phages from the same group (Bondy-Denomy et al, 2016; Tsao et al, 2018) by altering the O-antigen production (*gp9*), affecting bacterial twitching motility (*gp4*), and blocking phage genome injection (*gp30*) (Tsao et al, 2018) (Appendix Table S1). To evade host cell immunity, phage JBD30 encodes a gene inactivating the CRISPR/ Cas system (*gp35* anti-CRISPR protein AcrF1) (Appendix Table S1) (Bondy-Denomy et al, 2013). The *gp35* blocks the DNA-binding activity of the CRISPR-Cas by binding the Csy surveillance complex (Bondy-Denomy et al, 2015; Chowdhury et al, 2017). Moreover, *gp5* encodes the Aqs1 protein (quorum sensing anti-activator 1) that binds and inhibits LasR (Shah et al, 2021), which is a key protein in the regulation of *P. aeruginosa* quorum sensing (Appendix Table S1) (Juhas et al, 2005). By blocking LasR, the phage attenuates cell to cell communication, and the bacterial population is left vulnerable to phages. Besides this, Aqs1 binds to PilB, the ATPase that provides the energy necessary for pilus formation (Burrows, 2012), and thus blocks pilus assembly and further infection by phages requiring pili (Shah et al, 2021). Type IV pili are responsible for the twitching motility of *P. aeruginosa*, enable adhesion of the cells to surfaces, and play a role in biofilm formation (Burrows, 2012; Craig et al, 2019). Phages bind along the length of the pilus, and it was

Central European Institute of Technology, Masaryk University, Brno, Czech Republic. ✉E-mail: pavel.plevka@ceitec.muni.cz

speculated that they are pulled towards the cell surface by pilus retraction (Bradley and Pitt, 1974; Bradley, 1972). It was shown that JBD30 cannot infect cells that lack type IV pili (pilA deletion *P. aeruginosa* PA14 mutants) (Bondy-Denomy et al, 2016) nor cells that protect their pili by glycosylation (Harvey et al, 2017).

Despite extensive studies of the ability of JBD30 to evade the bacterial immune system (Bondy-Denomy et al, 2013, 2015; Chowdhury et al, 2017), attenuate host cell population (Shah et al, 2021), and the characterization of JBD30 moron genes (Bondy-Denomy et al, 2016; Tsao et al, 2018), its virion structure and the mechanism of infection have remained unknown. Here we present the structures of the virion and empty particle of siphophage JBD30 resolved using cryo-electron microscopy. The combination of cryo-electron tomography and fluorescent microscopy enabled us to describe the JBD30 infection cycle starting with binding to the type IV pilus, genome ejection into the host cell, assembly of progeny virions, and terminating with the release of new phages by cell lysis. Bacteriophage JBD30 belongs to the large group of siphophages that infect Gram-negative bacteria (Bondy-Denomy et al, 2016). The structural features and replication mechanisms described for bacteriophage JBD30 are likely shared by siphophages that utilize the type IV pilus for initial host cell recognition.

## Results and discussion

### Virion structure of phage JBD30

The virion of bacteriophage JBD30 is composed of an icosahedral capsid, long flexible non-contractile tail, and a baseplate (Fig. 1AB; Appendix Figs. S1 and S2; Appendix Table S2). The isometric capsid of JBD30 is formed by major capsid proteins (*gp38*) and decorated with trimers of minor capsid protein (*gp39*) (Fig. 1A–D; Appendix Fig. S3). A pentamer of major capsid proteins on one of the capsid vertices is replaced by a connector complex formed by twelve copies of each portal (*gp32*) and adaptor (*gp41*) proteins and a hexamer of stopper proteins (gp42) (Fig. 1B,D). The hexamer of stopper proteins provides an interface for tail attachment. The tail of bacteriophage JBD30 is 180 nm long and is built from 45 discs of major tail protein (*gp44*) hexamers (Fig. 1A,B,D). The disc of major tail proteins distal from the head binds to a trimer of distal tail proteins (*gp49*), which mediates a symmetry mismatch between the sixfold symmetry of the tail and threefold symmetry of the baseplate (Fig. 1B,D). The tail channel is filled with a trimer of tape measure proteins (*gp46*) whose C-termini extend into the baseplate core (Fig. 1B,D). The core of the baseplate is formed of a trimer of baseplate hub proteins (*gp53*) stabilized by three upper baseplate proteins (*gp50*) (Fig. 1B,D). The upper baseplate proteins together with the distal tail proteins form attachment sites for the tail fibres composed of three *gp47* and *gp48* heterodimers (Fig. 1B,D). The baseplate is terminated by three trimers of receptor-binding proteins (*gp54*) that form a tripod (Fig. 1B,D).

### JBD30 capsid is stabilized by minor capsid proteins

The icosahedral capsid of bacteriophage JBD30 is built from 415 copies of major capsid proteins organised in a T7 *laevo* lattice (Fig. 2A,B; Appendix Figs. S1 and S2, Appendix Table S2). The

classification, performed as a part of the single-particle reconstruction, identified two classes of JBD30 capsids. One of them contained minor capsid proteins whereas the other lacked them. The two types of particles were present in a ratio of 5:7 (Fig. 2A). Capsids from the first class contain trimers of minor capsid proteins attached around threefold and pseudo-threefold symmetry axes (Fig. 2A,D). The major capsid protein of JBD30 has the canonical HK-97 fold common to tailed bacteriophages and many other viruses (Fig. 2B,E) (Duda and Teschke, 2019; Suhanovsky and Teschke, 2015). The major capsid protein can be divided into the N-terminal arm, axial domain, peripheral domain, extended loop, and axial loops (Fig. 2E). The JBD30 major capsid protein is stabilized by a disulfide bond between Cys130 in the axial loop and Cys217 from the axial domain (Fig. 2F).

The capsid of JBD30 is formed by pentamers and hexamers of major capsid proteins (Fig. 2B). Contacts between the pentamers and hexamers are stabilized by the interactions of extended loops, peripheral domains, and N-terminal arms (Fig. 2B). The extended loop of the major capsid protein stretches over the peripheral domain of the neighbouring subunit positioned clockwise in the same hexamer or pentamer (Fig. 2B). Unlike in phage HK97, there are no covalent bonds linking the capsid proteins to each other (Wikoff et al, 2000). The three peripheral domains around threefold and pseudo-threefold axes connect either three hexamers, or two hexamers and one pentamer via interactions of α5 helices to form a phenylalanine lock (Phe192) (Fig. 2G). The N-terminal arm extends from the major capsid protein core, protrudes towards the neighbouring capsomer, and forms a small hook that reaches above the spine α-helix (Fig. 2B). On each pseudo-twofold axis, two N-terminal arm hooks interact with each other (Fig. 2C).

Differences between the capsid proteins forming a pentamer and hexamer are in the positioning of the N-terminal arm hook and extended loop (Fig. 2E). In hexamers, the N-terminal hook and extended loop are bent 22° and 25° upwards compared to those of the forming pentamers. The positioning of annular loops extending from the axial domains leaves openings in the centre of hexamers (7 Å radius) and pentamers (6 Å radius) (Fig. 2B). These pores were suggested to serve for water and ion passage during genome packaging and ejection (Selivanovitch et al, 2021), and putative scaffold protein release during capsid maturation (Chen et al, 2011; Guo et al, 2014).

Approximately one-half of phage JBD30 virions in our sample had capsids decorated with trimers of minor capsid proteins (*gp39*) (Fig. 2A). Minor capsid proteins of other phages have been shown to: stabilize the capsid architecture against internal genome pressure (T4 Soc, YSD1 *gp16*) (Qin et al, 2010; Hardy et al, 2020), help to increase the capsid size and volume (P23-45 *gp88*) (Bayfield et al, 2019), play a role in target cell recognition (T5 pb10, T4 Hoc) (Vernhes et al, 2017; Sathaliyawala et al, 2010), and participate in virion assembly (λ *gpD*) (Lander et al, 2008). Similarly to auxiliary proteins of phage λ (*gpD*) (Lander et al, 2008), phage YSD1 (*gp16*) (Hardy et al, 2020), and phage P23-45 (*gp88*) (Bayfield et al, 2019), the minor capsid proteins of bacteriophage JBD30 form homotrimers docked on the threefold and pseudo-threefold axes (Fig. 2A). Whereas the auxiliary proteins of phages λ, YSD1, and P23-45 form an outer chainmail net with the trimers connected to each other by their N-terminal hook extensions (Hardy et al, 2020; Bayfield et al, 2019; Lander et al, 2008), the minor capsid protein trimers of JBD30 are not connected (Fig. 2A).

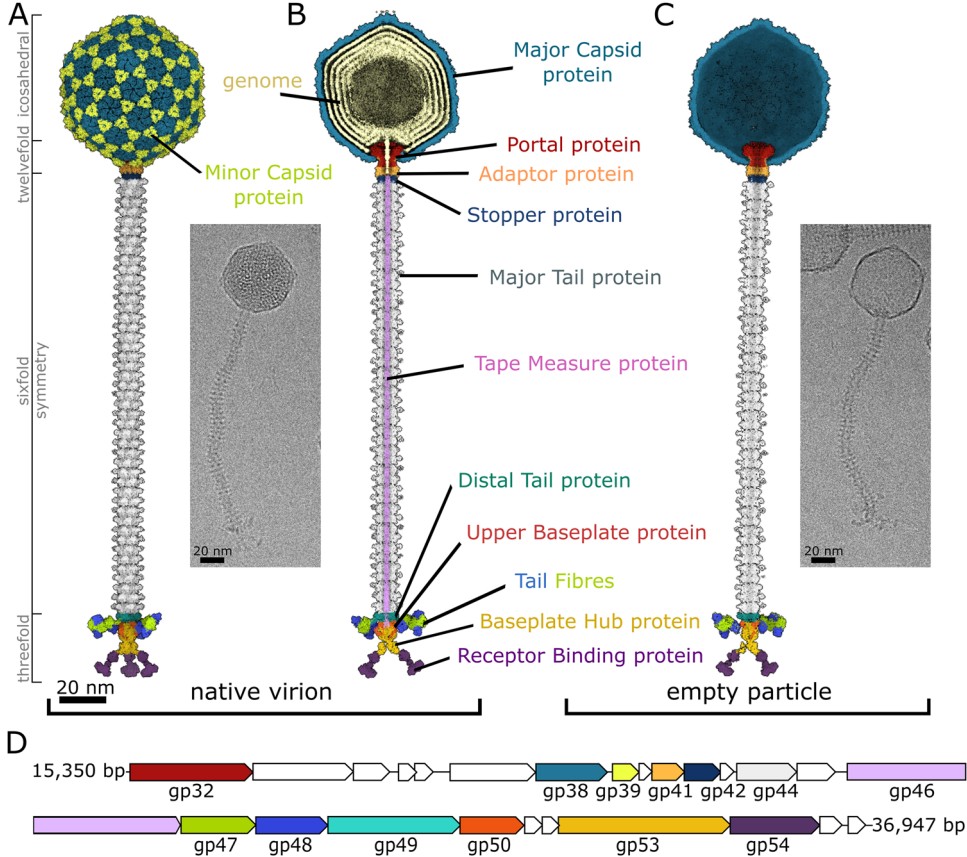

**Figure 1. Structure of bacteriophage JBD30 and its empty particle.**

Surface representation of composite cryo-EM map of JBD30 (**A**) virion, (**B**) virion with front half removed, and (**C**) empty particle with front half removed. Maps are coloured according to the components occupying the density. Blue—major capsid protein (*gp38*), lime—minor capsid protein (*gp39*), ochre—dsDNA genome, maroon—portal protein (*gp32*), apricot—adaptor protein (*gp41*), navy blue—stopper protein (*gp42*), grey—major tail protein (*gp44*), light violet—tape measure protein (*gp46*), turquoise—distal tail protein (*gp49*), fire brick red—upper baseplate protein (*gp50*), green—tail fibre forming protein *gp47*, ultramarine blue—tail fibre forming protein *gp48*, saffron—baseplate hub protein (*gp53*), plum—receptor binding protein (*gp54*). Insets show electron micrographs of JBD30. (**D**) Scheme of JBD30 genome segment encoding structural genes. The genes are coloured the same as their protein products in panels (**A–C**). Open reading frames coloured in white are non-structural genes.

The fold of minor capsid protein of JBD30 consists of a three-stranded antiparallel β-sheet, short β-hairpin stabilized by a disulphide bond between Cys67 and Cys72, and an α-helix (Fig. 2D). Cryo-EM density of the N-terminal region of the minor capsid protein, residues 1–52, was not resolved enough to enable determination of its structure. The interactions of the minor capsid proteins with major capsid proteins are mainly electrostatic. Negatively charged minor capsid protein residues Asp96, Ser95, and Glu91 form a pocket for Lys66 from the extended loop of the major capsid protein (Fig. 2H). Lys197 from α5 helix of the peripheral domain of the major capsid protein forms a salt bridge with the minor capsid protein residue Glu198 (Fig. 2H). The major capsid protein residues Glu198, Ser194 and Thr193 interact with the minor capsid protein residue Lys100 (Fig. 2I).

Despite low sequence identity (13.1%), the topology of the JBD30 minor capsid protein resembles the β-tadpole fold of the T4 auxiliary protein Soc (Qin et al, 2010; Dedeo et al, 2020). The β-tadpole fold consists of a tadpole head formed of a three-stranded β-sheet and two α-helices, and a tadpole tail formed of a β-hairpin (Fig. 2D) (Qin et al, 2010). While the tadpole tail of T4 Soc extends straight from the three-stranded β-sheet, that of JBD30 *gp39* is bent

towards it (Fig. 2D). Phage T4 was shown to be stable without the Soc trimers (Ishii and Yanagida, 1977), nevertheless, the presence of Soc trimers increases the chance of T4 withstanding higher temperatures (increase in denaturation temperature by 6 °C) (Ross et al, 1985) and extremes of pH ($10^4$-fold improved survival at pH 10.6) (Ishii and Yanagida, 1977). We hypothesise that the minor capsid protein is not vital for JBD30 under mild conditions. Similarly to Soc of T4, the minor capsid protein of JBD30 could enable the phage to withstand unfavourable conditions. However, the minor capsid protein is likely to be dispensable for capsid morphogenesis and phage infection.

## Genome organization

The genome of phage JBD30 consists of 36,947-bp-long dsDNA and contains 56 open reading frames (Bondy-Denomy et al, 2013). Its high GC content (64.3%) corresponds to that of its host bacterium *P. aeruginosa* (Klockgether et al, 2011; Kiewitz and Tümmler, 2000). Bacteriophage JBD30 belongs to phages that employ headful genome packaging mechanism (George and Bukhari, 1981). The asymmetric reconstruction of the JBD30 head

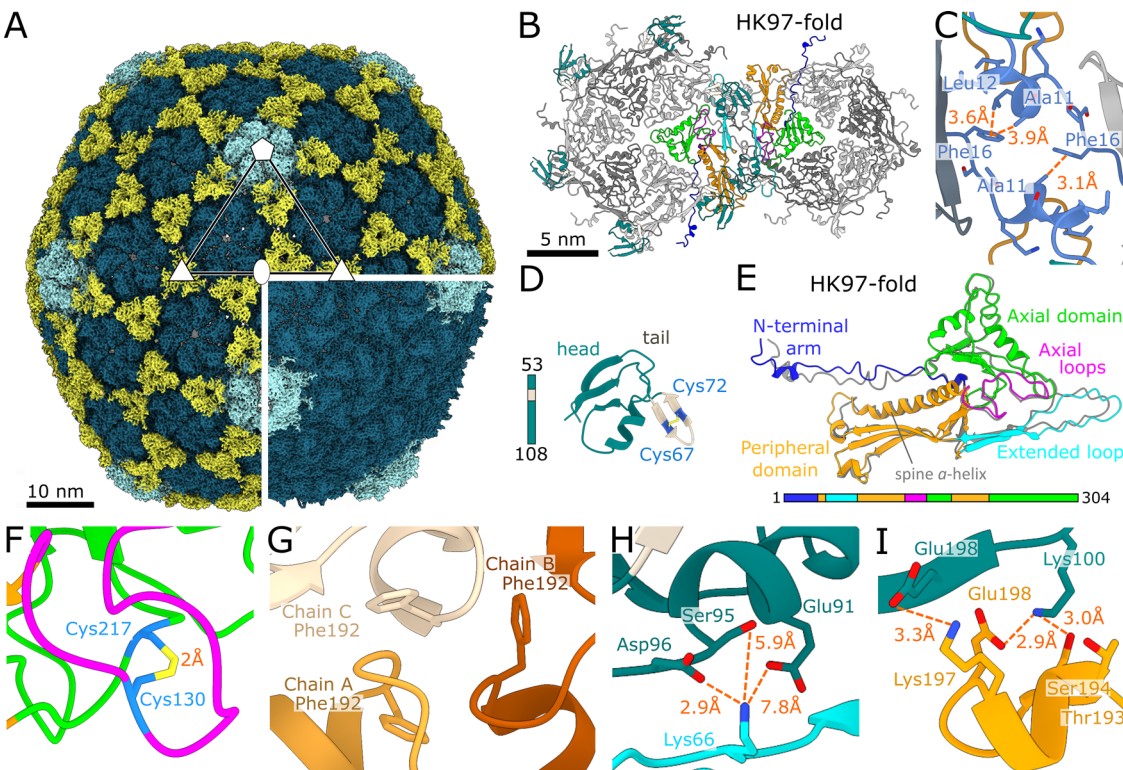

**Figure 2. Structure of JBD30 capsid.**

(A) Molecular surface representation of JBD30 capsid reconstructed with imposed icosahedral symmetry. Density occupied by pentamers of major capsid protein is highlighted in light blue, hexamers in turquoise, and minor capsid protein trimers in yellow. The positions of selected icosahedral five-, three-, and twofold symmetry axes are indicated. The bottom right quarter shows the surface of JBD30 capsid without the minor capsid proteins. (B) Cartoon representation of hexamer and pentamer of major capsid proteins. Two interacting major capsid proteins are coloured according to domains as in panel (E). Minor capsid proteins are coloured as in panel (D). The other proteins are shown in alternating light and dark grey. (C) Cartoon representation of the interaction of major capsid proteins N-terminal hooks around twofold icosahedral axis of the capsid. (D) Cartoon representation of JBD30 minor capsid protein coloured according to domain composition and viewed from the capsid. Cysteines forming a disulphide bond in the tadpole tail are shown in stick representation. (E) Cartoon representation of JBD30 major capsid protein forming a hexamer. Individual domains are coloured according to the sequence diagram shown at the bottom of the panel. The structure of the major capsid protein forming a pentamer is superimposed onto that forming a hexamer and shown in grey. (F) Detail of a disulphide bond between the Cys130 from the axial loop of the major capsid protein and Cys217 from the axial domain. The major capsid protein is coloured as in panel (E). (G) Detail of phenylalanine lock between three adjacent peripheral domains of major capsid proteins. Chain C and chain B are hexamer-forming major capsid proteins, chain A belongs to a pentamer. (H) Minor capsid protein residues Glu91, Ser95, and Asp96 (in dark blue) interact with Lys66 from the extended loop of another major capsid protein (cyan). Selected interatomic distances are indicated as dark orange dashed lines. (I) Interactions of the major capsid protein peripheral domain helix α5 (orange) with the minor capsid protein (blue-green). Glu198 and Lys100 of the minor capsid protein form salt bridges with residues from the adjacent major capsid protein.

contains five resolved concentric layers of genome density (Fig. 1B; Appendix Fig. S4A–C). The outer-most genome density shell is separated into strands that wind underneath the capsid (Appendix Fig. S4A). In addition, the reconstruction contains a ring of DNA density with resolved minor and major grooves above the wing domains of the portal complex (Appendix Fig. S4A,D). The closed ring structure of the DNA segment is an artifact of the cryo-EM reconstruction that originates from the averaging of the information from projection micrographs of numerous phage heads.

## Portal complex

One of the pentamers of major capsid proteins that form the JBD30 capsid is replaced by a connector complex, which consists of a portal protein dodecamer, adaptor protein dodecamer, and stopper protein hexamer (Fig. 3A). The JBD30 portal protein

(gp32) has the same domain organization as portals of other dsDNA phages, such as T4 (Fang et al, 2020), SPP1 (Orlov et al, 2022), and T7 (Cuervo et al, 2019). It is composed of crown, wing, stem, and clip domains (Fig. 3A). The wing domain of the JBD30 portal protein consists of six α-helices framed by peripheral β-strands and flexible loops (Fig. 3B). Residues Arg13, Arg18, and Arg146 form a positively charged rim in the upper part of the wing domain that interacts with a segment of DNA wrapping around the portal dodecamer (Appendix Figs. S4A,D,E and S5C). The second positively charged rim is formed by the twelvefold repeated residues Lys44, Lys38, and Arg47, which interact with the inner face of the capsid (Appendix Fig. S5C). Two α-helices of the portal protein stem domain cross the capsid shell, and together with the clip domain provide an interface for the attachment of adaptor proteins (Fig. 3A–C). A loop from the clip domain intertwines between two neighbouring portal proteins and stabilizes the portal complex (Fig. 3A–C).

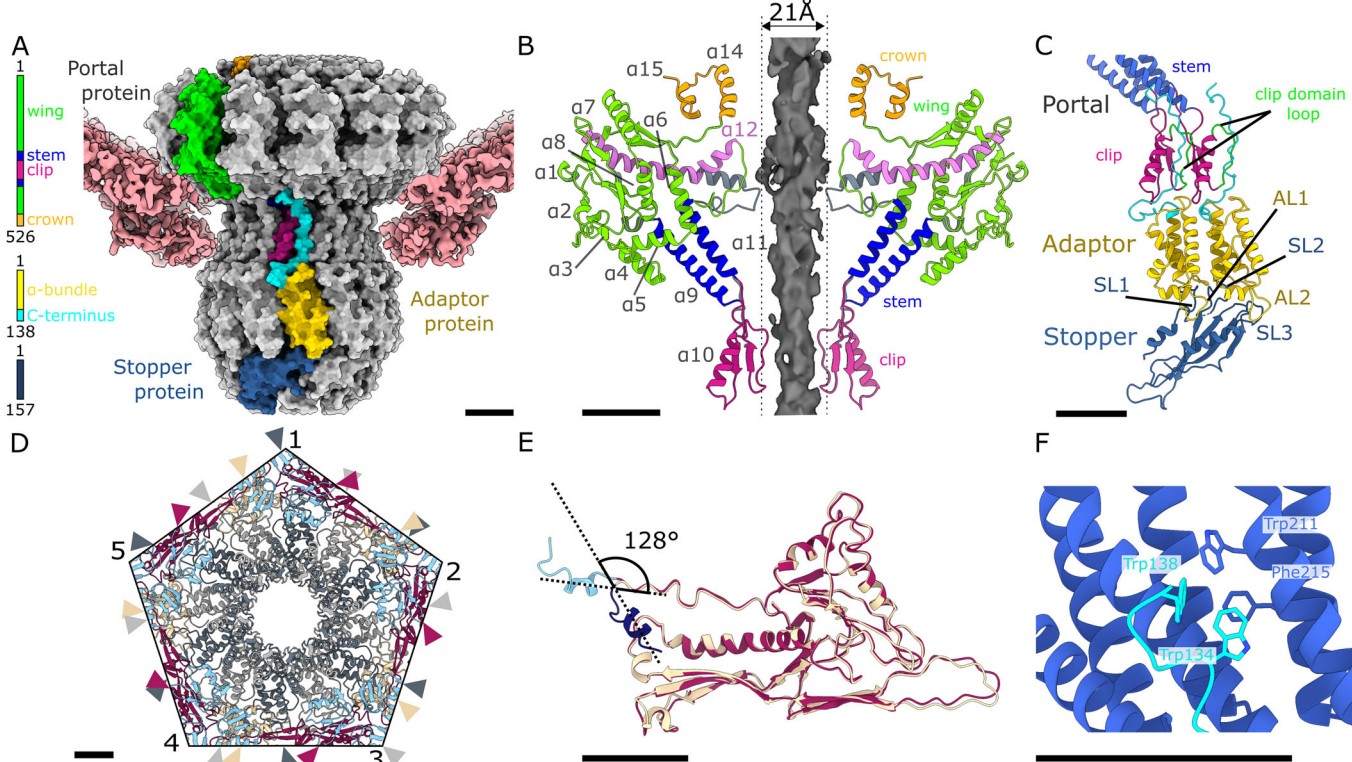

**Figure 3. Structure of JBD30 connector complex.**

(A) Molecular surface representation of JBD30 connector composed of a dodecamer of portal proteins, dodecamer of adaptor proteins, and hexamer of stopper proteins. The cryo-EM density of capsid proteins surrounding the connector complex is shown in surface representation in pink. One subunit of each of the connector proteins is coloured according to the domain diagrams on the left of the panel. (B) Portals of virion and empty particle differ in the structure of wing domain helix α12. Cartoon representation of two opposite portal proteins from a virion coloured according to domains (as in panel (A)). Portal proteins from an empty particle are depicted in grey. Helix α12 is fully stretched in the empty particle and the tunnel loop narrows the portal channel to 21 Å. (C) Symmetry reduction from twelvefold of the adaptor and sixfold of the stopper complex. The α-helical bundles of two adjacent adaptor proteins sit on stopper protein loops SL1, SL2, and SL3. Loops AL1 and AL2 of the adaptor protein reach over the stopper protein rim. The C-terminus of the adaptor protein runs through the cleft between the clip domains of two adjacent portal proteins. Cartoon representations of portal, adaptor, and stopper proteins are coloured according to domains as in panel (A). The clip domain loops that intertwine between the neighbouring portal proteins are highlighted in green. (D) Symmetry mismatch between capsid and portal. The capsid–portal interface is viewed along the tail axis from the outside of the particle. The portal proteins are shown in alternating grey and dark grey, the major capsid proteins are coloured in alternating purple and beige, the minor capsid proteins are shown in light blue. Positions of individual portal and major capsid proteins are indicated by arrows. (E) Superimposition of cartoon representations of major capsid protein interacting with another capsomer (beige and N-terminus in dark blue) and capsid protein adjacent to portal dodecamer (purple and N-terminus in light blue) and one showing the bending of the N-terminal arm which enables incorporation of the portal complex into the capsid. (F) Interactions between portal protein stem domain (blue) and adaptor protein C-terminal hook (cyan). Side chains of the interacting residues are shown in stick representations. Scale bars represent 25 Å.

The asymmetric reconstruction of the capsid–portal interface shows the symmetry mismatch between fivefold symmetry of the capsid and twelvefold symmetry of the portal (Fig. 3D). The portal complex is surrounded by five hexamers of major capsid proteins. Two major capsid proteins from each of the hexamers contact the portal (Fig. 3D,E). One of the major capsid proteins binds the portal complex with its extended loop and peripheral domain. The second major capsid protein, which is positioned counter-clockwise when looking at the portal from the inside of the capsid, contacts the portal with its peripheral domain. The N-termini of major capsid proteins that interact with each other point towards the pseudo-twofold axes between two neighbouring capsomers (Fig. 2B). In contrast, at the interface with the portal complex the N-terminus of the major capsid protein (residues 2–14) turns back and forms a short α-helix positioned parallel to the extended loop of the neighbouring major capsid protein, thus leaving more space around the fivefold axis for the portal complex (Fig. 3E).

## Head completion proteins

The portal complex is followed by head completion proteins: a dodecamer of adaptor proteins and hexamer of stopper proteins (Fig. 3A). The adaptor protein (*gp41*) is formed by an α-helical bundle stabilized by a hydrophobic core and an extended C-terminus (Fig. 3C). The C-terminus wedges into the grove between the clip domains of two adjacent portal proteins, continues along the stem domain α-helices, and ends at the beginning of the wing domain helix α4 (Fig. 3A–C). The interaction between the portal and adaptor proteins is stabilized by π–π stacking between Trp134 and Trp138 of the adaptor protein and Phe215 and Trp211 of the portal protein (Fig. 3F). The reduction of symmetry from the twelvefold of the connector to the sixfold of the tail occurs at the adaptor–stopper interface (Fig. 3A,C). Each stopper protein (*gp42*) interacts with two adaptor proteins. The interactions between adaptor and stopper proteins are mainly electrostatic (Appendix

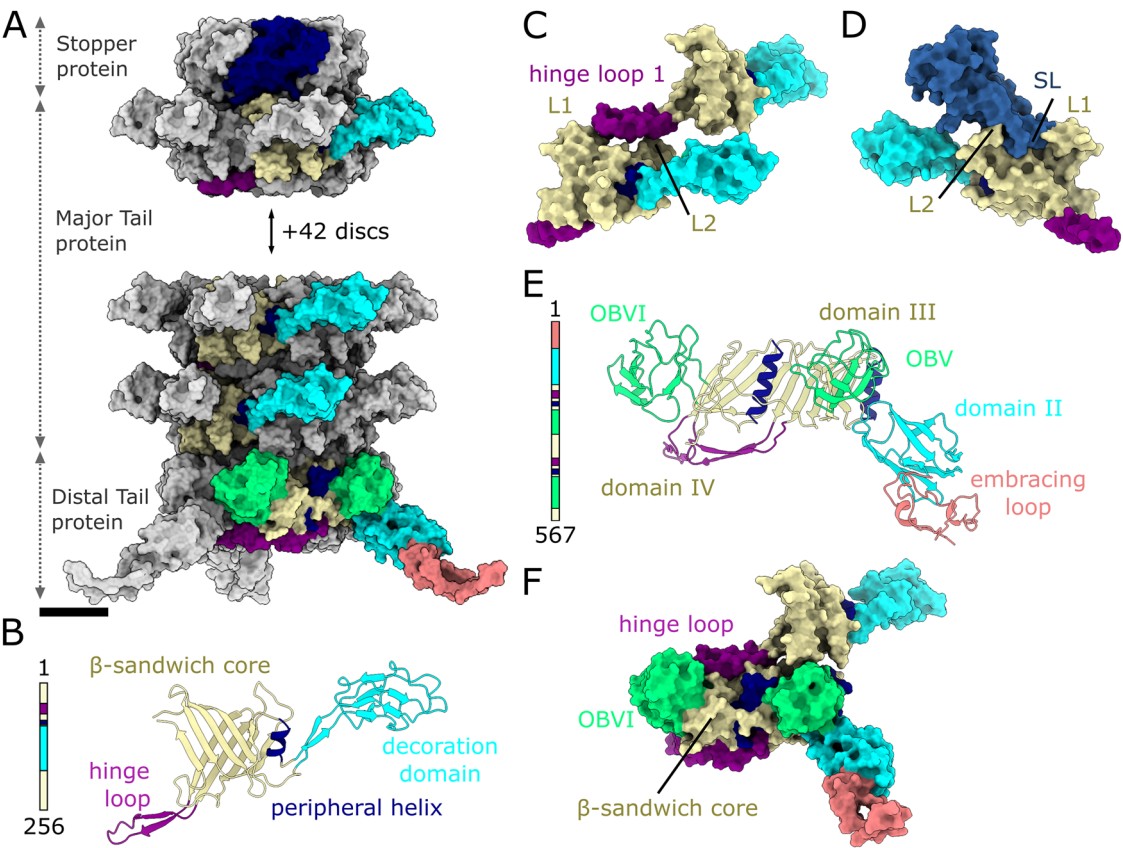

**Figure 4. Structure of JBD30 tail.**

(A) JBD30 tail attaches to hexamer of stopper proteins and is formed of 45 hexamers of major tail proteins, and trimer of distal tail proteins. The proteins are shown in molecular surface representation. One stopper protein is highlighted in dark blue. Selected subunits of the major tail protein and distal tail protein are coloured according to domains as in panels (B) and (E). Each subunit of the distal tail protein contains two β-sandwich domains and thus enables the reduction of symmetry from sixfold of the tail to threefold of the baseplate. (B) Cartoon representation of major tail protein with domains highlighted in colour according to the sequence diagram on the left of the panel. (C) Detail of contacts between subunits from successive discs of major tail protein. Molecular surface representations of the two major tail proteins are coloured according to domains as in panel (B). The hinge loop 1 from the top major tail protein fits into the space between bottom major tail protein loops L1 (residues 194–198) and L2 (residues 217–227). (D) Molecular surface representation of stopper protein–major tail protein interaction. The stopper protein (blue) long loop stretches above the major tail protein and fits between loops L1 and L2. The major tail protein is coloured according to domains as in panel (B). (E) Cartoon representation of distal tail protein domain composition coloured according to the sequence diagram on the left of the panel. OB domain stands for oligosaccharide binding domain. (F) Interaction between major tail protein and distal tail protein. The major tail protein hinge loop is positioned on top of distal tail protein β-sandwich core. The tip of the hinge loop ends behind the distal tail protein OB domain VI.

Fig. S5). Whereas the surface charge of the adaptor interface is positive, the interacting residues of the stopper protein are mostly negatively charged (Appendix Fig. S5). The stopper protein is composed of a four-stranded β-sheet and two α-helices connected by flexible loops (Fig. 3C). Homologues of the stopper protein are found in the head-to-tail joining interface of the *Rhodobacter capsulatus* gene transfer agent, bacteriophage SPP1, and bacteriophage λ (Appendix Table S2) (Bárdy et al, 2020). Whereas the neck regions of bacteriophages SPP1 and λ are formed of three consecutive rings of adaptor, stopper, and tail-to-head joining proteins (*gp15*, *gp16*, *gp17* for SPP1 (Lhuillier et al, 2009) and *gpW* (Maxwell et al, 2001), *gpFII* (Maxwell et al, 2002), *gpU* (Edmonds et al, 2007) for phage λ), the neck of bacteriophage JBD30 has only adaptor and stopper proteins. The stopper proteins of JBD30 form an interface for tail attachment (Figs. 1B and 4A,D).

## Interaction of connector complex with DNA

The diameter of the portal channel at its narrowest is 30 Å in a virion and 21 Å in an empty JBD30 particle (Figs. 1A,C and 3B). The difference is caused by the conformational change of helices α12 from the wing domains and tunnel loops of the portal proteins (Fig. 3B). The two distinct portal conformations: opened and closed, were previously described for bacteriophages T7 and P23-45, in which the opening and closing of the tunnel loop valve were proposed to control the DNA encapsidation and exit (Bayfield et al, 2019; Cuervo et al, 2019).

Asymmetric reconstruction of the JBD30 virion connector complex contains double-stranded DNA passing through the portal channel (Appendix Fig. S4C–E). The DNA does not interact with the stopper proteins and continues uninterrupted into the tail discs (Appendix Fig. S4D,E). It is likely that deeper in the tail channel the

end of the DNA is held in place by interactions with a trimer of tape measure proteins. The tunnel loops of the JBD30 portal proteins interact with DNA in the channel (Appendix Fig. S4E). Whereas most of the portal protein structure exhibits twelvefold symmetry, the structures of tunnel loops (residues 301–LGGTLTSTTSQSGGGAFALGQVHNE–325) differ between the individual subunits of the complex (Appendix Fig. S4F). The Phe317 residues of three portal proteins bind to one side of the double-stranded DNA in the channel, suggesting a π–π interaction between the amino acid side chains and riboses of the DNA (Appendix Fig. S4F). This interaction could stabilize the DNA inside the phage capsid. The DNA does not interact with adaptor or stopper proteins.

## Tail tube

The most prominent feature of siphophages is their long flexible non-contractile tail. The tail of bacteriophage JBD30 is a right-handed six-entry helix composed of 45 discs, each built from six major tail proteins (*gp44*) (Figs. 1A,B and 4A). The twist and rise in between two adjacent discs are 23.5° and 43.2 Å, respectively (Fig. 4A).

The major tail protein of JBD30 consists of a central β-sandwich (residues 8–41, 62–73, 184–256), peripheral α-helix (residues 77–83), decoration domain (residues 88–175), and a long hinge loop (residues 42–62) (Fig. 4B). The inner tail tube diameter, 40 Å, is defined by the β-barrel formed by 24 β-strands, to which each major tail protein contributes four β-strands. The interior of the tube is smooth and round in cross-section (Appendix Fig. S5). The surface is negatively charged to enable the swift passage of the DNA (Appendix Fig. S5). As in the tails of phages T5 (Arnaud et al, 2017) and λ (Pell et al, 2010), the decoration domains of JBD30 major tail protein have an immunoglobulin-like fold, and protrude tangentially from the major tail protein disc (Fig. 4A,B). The disc-to-disc contacts are mediated by the hinge loops (Fig. 4B,C), each of which is inserted between loops formed by residues 217–227 of two major tail proteins from an adjacent hexamer (Fig. 4C). We assume that the flexibility of the hinge loop and its interaction with the loops of adjacent major tail proteins enables the bending of the tail, similar to those of phages SPP1 (Zinke et al, 2020) and λ (Campbell et al, 2020).

The JBD30 tail contains a long density inside the tail tube, which was absent in the tail tube reconstructed from empty particles (Fig. 1B,C). The density is continuous and ~20 Å in diameter, but it is not resolved enough to enable structure determination. We speculate that this density belongs to a trimer of tape measure protein (*gp46*).

## Structure of JBD30 baseplate

Reconstructions of the whole JBD30 baseplate and its parts enabled us to assemble a composite map that combines the localized reconstructions of the baseplate core, receptor binding proteins, and tail fibres (Fig. 5A). The bacteriophage JBD30 baseplate is composed of a trimer of distal tail proteins, three tail fibres—each composed of three heterodimers of *gp47* and *gp48* subunits, three upper baseplate proteins, a trimer of baseplate hub proteins, and three trimers of receptor binding proteins (Fig. 5A).

The trimer of distal tail proteins (*gp49*) forms a ring that binds to the distal disc of major tail proteins (Fig. 4A,E,F). The distal tail protein consists of six domains (Fig. 4E). The N-terminal domain I (residues 2–74) forms a long loop that wraps around the tail fibres (Fig. 5A). Domain II (residues 75–174) stretches in a perpendicular direction from the tail axis (Figs. 4A and 5A). Domain III (residues 175–251, 321–356) and domain IV (residues 357–443, 533–567) are positioned next to each other around the tail axis, and together form an interface that enables the binding of one distal tail protein to two major tail proteins (Fig. 4A,F). Both domains have the same fold, with cores formed by β-sandwiches flanked by α-helices, thus mimicking the organization of the major tail protein (Fig. 4B,E). Both domains III and IV are decorated with one of oligosaccharide binding domains V and VI (residues 252–320 and 444–532, respectively) (Fig. 4E).

The core of the baseplate is formed of a trimer of baseplate hub proteins (*gp53*) (Fig. 5A). The shape of the baseplate hub protein trimer resembles a trophy cup, as it was termed for phage T5 (Linares et al, 2023) (Fig. 5A). The baseplate hub protein is composed of seven domains (Fig. 5B). The N-terminus starts at the bottom of the cup and continues upwards to domains I, II, and III (Fig. 5B). Domain IV forms an interface for binding to distal tail proteins and gives rise to a long linker, which spans the whole baseplate hub monomer (Fig. 5B). This linker separates neighbouring baseplate hub proteins, continues to the baseplate tip, and ends in domain V. Domains V and VII have fibronectin-like folds (Fig. 5B). The small, six-stranded β-barrel domain VI extends from domain V and contacts domain VII, thus stabilising the baseplate architecture (Fig. 5B). The tip of the baseplate is closed by three α-helices, each protruding from one baseplate hub domain V (Fig. 5B; Appendix Fig. S6). Besides playing the role of a baseplate organiser, the trimer of baseplate hub proteins also provides attachment sites for the tail fibres and receptor-binding proteins.

The upper baseplate protein (*gp50*) is composed of two domains, each formed of a central β-sandwich with an α-helix, connected by a flexible linker (Fig. 5A; Appendix Fig. S6E). Domain I of the upper baseplate protein sits in a gap between the domain IVs of two baseplate hub proteins (Appendix Fig. S6F). The upper baseplate protein domain II is attached to the surface of the baseplate hub domain III (Fig. 5A; Appendix Fig. S6E,F). Residues Cys147, Cys156, Cys238, and Cys245 coordinate a putative metal ion in domain II of the upper baseplate protein (Fig. 5D). This metal–sulphur cluster can be important for baseplate stability, as was described for iron-sulphur clusters in baseplate hub proteins of *Rhodobacter capsulatus* gene transfer agent (Bárdy et al, 2020) and phage λ protein *gpL* (Tam et al, 2013). The linker runs from upper baseplate protein domain I to the baseplate interior, loops around adjacent baseplate hub domain II and returns to the outer baseplate surface, where it continues into upper baseplate protein domain II (Fig. 5B; Appendix Fig. S6E,F). The upper part of the upper baseplate protein linker is negatively charged, and thus complementary to the positively charged surface of baseplate hub protein domain II (Appendix Fig. S5). The extended C-terminus of the upper baseplate protein is buried inside the baseplate core between two domains I of baseplate hub proteins (Fig. 5A). The interaction between the baseplate hub proteins, upper baseplate proteins, and distal tail proteins is mediated by the long loops of distal tail protein (residues 193–214, 387–408), loops from domain I (residues 8–14, 32–51) of the upper baseplate protein, and loop of domain IV (residues 371–377) of the baseplate hub protein (Fig. 5A). The highly electronegative surface of the baseplate

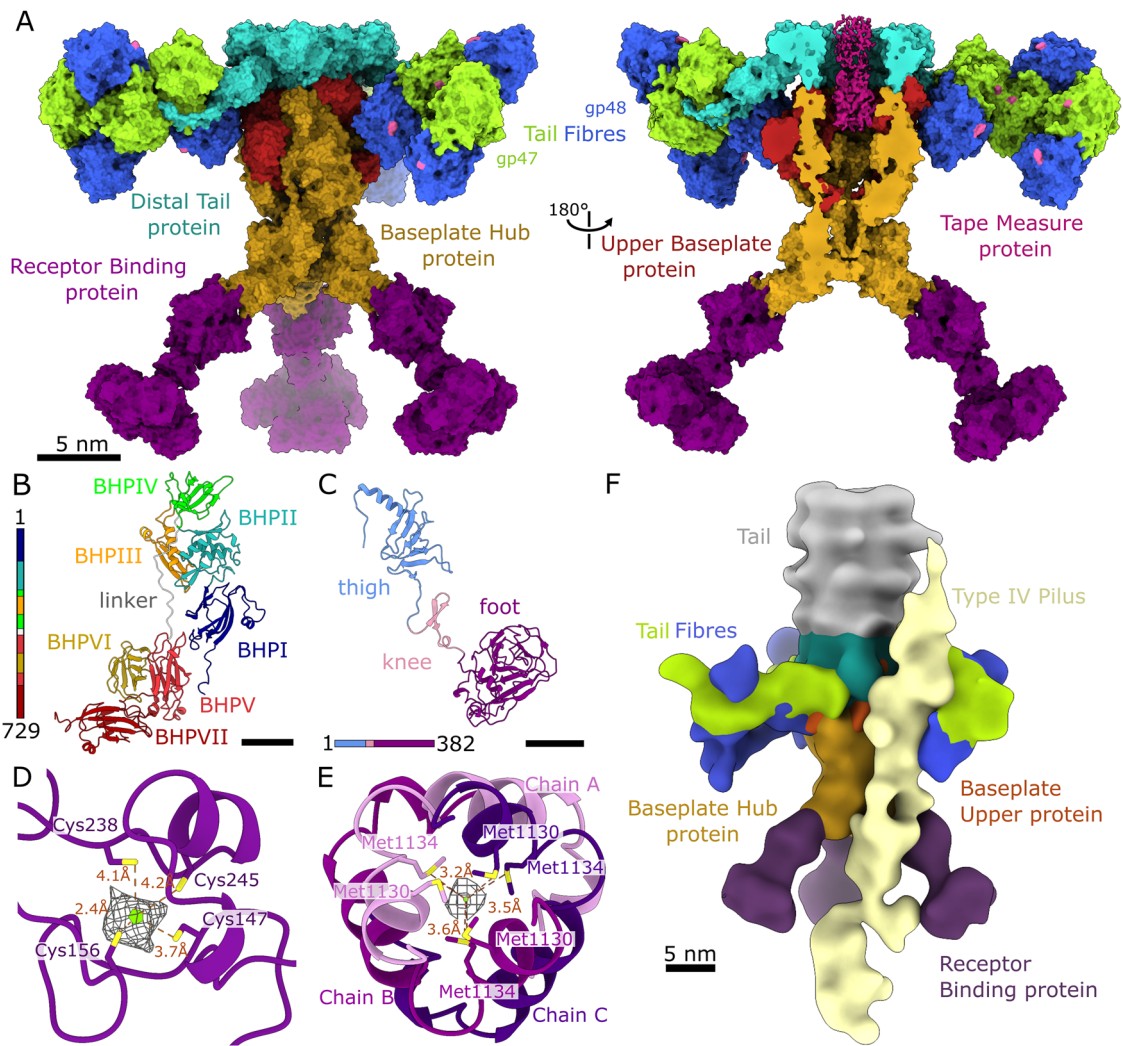

**Figure 5. Structure of JBD30 baseplate.**

(A) Side views of molecular surface representation of JBD30 baseplate and baseplate with front half removed. The distal tail protein is shown in cyan, baseplate hub protein in saffron, receptor binding protein in plum, upper baseplate protein in fire brick red, tail fibre forming protein *gp47* in green, tail fibre forming protein *gp48* in ultramarine blue, and the cryo-EM density occupied by the tape measure protein is highlighted in magenta. Pilus interaction interfaces of *gp47* and *gp48* are highlighted in pink. (B) Cartoon representation of baseplate hub protein with domains coloured according to the sequence diagram on the left of the panel. (C) Cartoon representation of receptor binding protein with domains coloured according to the sequence diagram at the bottom of the panel. Scale bars 25 Å. (D) Detail of putative metal binding site in upper baseplate protein domain II. The distances between the sulphur atoms of cysteine side chains (yellow) and putative metal ligand (green) are indicated. (E) C-terminal domain of tape measure protein trimer viewed along tail axis. The strong cryo-EM density between the three methionine residues that may belong to an ion is shown in mesh representation. Distances between the density centre and surrounding sulphur atoms of methionine residues are indicated. (F) Surface representation of cryo-EM reconstruction of JBD30 baseplate bound to the type IV pilus. The density is coloured according to the protein type as in panel (A), density of the type IV pilus is highlighted in pale yellow.

hub–upper baseplate protein complex provides a complementary charged interface for binding to the ring of distal tail proteins (Appendix Fig. S5).

Three tail fibres resembling bent arms protrude laterally from the baseplate (Fig. 5A). Each of the arms is composed of three heterodimers of *gp47* and *gp48*, which are rotated 180° relative to each other (Fig. 5A). The folds of *gp47* and *gp48* are similar (Appendix Fig. S6A; Appendix Table S3). *Gp47* was classified as a hydrolyse belonging to a nitrilase superfamily of proteins. This family includes nitrile- and amide-hydrolysing enzymes (Mistry et al, 2021). The cell wall of *P. aeruginosa* is composed of 15%

peptidoglycan and 30% proteins (Clarke et al, 1967). Thus, *gp47* may play a role in the cleavage of N-acetylglucosamine and peptides crosslinking the glycan strands (Vollmer and Bertsche, 2008). In contrast, *gp48* could not be classified into any enzyme family. A loop and three-stranded β-sheets of *gp47* form a positively charged surface depression, which binds the negatively charged α-helix of *gp48* (Appendix Fig. S6B). The *gp48* fold is stabilized by a disulfide bond between Cys142 and Cys174 (Appendix Fig. S6C).

Despite the only 13% amino acid sequence identity of JBD30 baseplate hub protein *gp53* and baseplate hub protein pb3 of

bacteriophage T5 (Linares et al, 2023) (Sievers et al, 2011), these proteins share the same fold and domain organization (Appendix Fig. S6F,G, Appendix Table S3). The baseplate hub protein of phage T5 consists of six domains and a long flexible linker (Appendix Fig. S6G). Whereas the core of T5 baseplate is formed of a single trimer of baseplate hub protein pb3, in JBD30 it is built from a combination of a baseplate hub trimer and three monomers of upper baseplate protein (Fig. 5A; Appendix Fig. S6F–H). The similarities between the T5 baseplate hub protein and JBD30 baseplate hub and upper baseplate proteins are extensive. We hypothesise that this general baseplate core architecture may be common to most, if not all, siphophages that infect gram-negative hosts.

The baseplate of JBD30 is terminated by a tripod formed by three trimers of receptor binding proteins (gp54) (Fig. 5A). The receptor binding protein can be divided into thigh, knee, and foot domains (Fig. 5C). The N-terminal thigh domain (residues 1–118) connects the receptor binding protein to domain VII of the baseplate hub protein (Fig. 5A). The C-terminus of the baseplate hub protein docks into the centre of an α-helical bundle that each receptor binding protein contributes one α-helix to (Appendix Fig. S6I). The α-bundle is surrounded by peripheral β-sheets. Structural homologues of the thigh domain are tail spikes of phages vB_ApiP_P1 and CBA120, and tail fibres of phage T7 (Holm, 2022). These proteins anchor the tail spike/fibre to the rest of the baseplate. The knee domain extends from the thigh domain, then bends back in an antiparallel β-sheet and ends in a short α-helix (Fig. 5C).

The C-terminal foot domain of the receptor binding protein has the overall shape of a wedge, which is composed of a central β-sheet flanked by an α-helical spur (Fig. 5C). The JBD30 receptor binding protein foot domain is structurally similar to sugar-binding domains of hydrolytic enzymes (Sainz-Polo et al, 2015) (such as xylanase, glycoside hydrolase, endoglucanase, mannose, galactosidase) and an enzymatically active domain of bacteriophage cba120 tail spike protein (Plattner et al, 2019). Aromatic residues of two adjacent foot domains form a hydrophobic cleft suitable for sugar binding (Appendix Fig. S6J). Therefore, we speculate that the foot domain of the JBD30 receptor binding protein recognizes and binds the host cell lipopolysaccharide O-antigen. The importance of the O-antigen for JBD30 binding is indirectly supported by the observation that a prophage JBD30 reduces host O-antigen length and thus prevents superinfection by other JBD30 phages (Tsao et al, 2018).

## Mechanism of JBD30 binding to the type IV pilus

The type IV pili serve as initial receptors of numerous phages infecting *P. aeruginosa* (Bondy-Denomy et al, 2016; Bradley and Pitt, 1974; Harvey et al, 2017; Chibeu et al, 2009; Bae and Cho, 2013; McCutcheon et al, 2018; Kim et al, 2012). Previous studies have shown phage binding along pili, which are helical structures formed by the major pilin protein *pilA* (Bradley and Pitt, 1974; McCutcheon et al, 2018). However, the mechanism of the interaction between the phage and the pilus remained unknown. We used cryo-EM to image JBD30 bound to the type IV pilus of *P. aeruginosa* and reconstructed the structure of a baseplate-pilus complex to the resolution of 26 Å (Fig. 5F). The low-resolution structure enabled the fitting of JBD30 baseplate and 50 nm long

segment of the *P. aeruginosa* type IV pilus (Wang et al, 2017). Bacteriophage JBD30 uses its baseplate tail fibre to bind the pilus (Fig. 5F). The arrangement of the three heterodimers of proteins *gp47* and *gp48* is such that they form a unique interface, which can embrace the pilus (Fig. 5A,F). Alphafold2 multimer (Evans et al, 2021) prediction of the interaction of *pilA* with *gp47* and *gp48* indicates that the JBD30 tail fibres bind the β-sheet rich region of *pilA* (Appendix Fig. S6K). Side chains of *pilA* residues Lys68 and Lys112 are predicted to interact with Asp185 of *gp47* and Tyr169 of *gp48*, respectively (Appendix Fig. S6K). It was shown previously that glycosylation of the *pilA* C-terminus, including the residue Lys112, blocks phage binding (Harvey et al, 2017). The receptor binding protein of JBD30 forms a putative contact with the type IV pilus (Fig. 5F). This interaction might help to stabilize the baseplate in an orientation along the pilus towards the bacterial cell surface. However, no interaction between the receptor binding protein and major pilin protein *pilA* was predicted using Alphafold2 multimer (Evans et al, 2021). The baseplate-pilus interaction orients the tripod of receptor-binding proteins of the baseplate towards the bacterial surface (Fig. 5F), which ensures that after reaching the cell surface, the baseplate is properly oriented for cell attachment and genome ejection. It is likely that other phages utilizing pili for cell binding employ a similar geometry of baseplate-pilus interactions.

JBD30 attachment to the type IV pilus enables two possibilities for how the phage can reach the cell surface. One-dimensional diffusion of a phage particle along the pilus may bring it to the cell surface. Alternatively, the phage may bind to one segment of the pilus and be carried to the cell surface by pilus retraction. To differentiate between the two alternatives, we added JBD30 to bacterial cells incubated at 4 °C. At this low temperature phage particles attached to pili, but most of them did not reach the cell surface (Appendix Fig. S7). After heating the cells to 37 °C, phage particles were brought to the cell surface. Lowering the temperature from 37 °C to 4 °C only causes a 10% reduction in the diffusion rate (Berg, 1993). However, cooling to 4 °C reduces *P. aeruginosa* metabolic activity and the pili polymerization and retraction dynamics (Tsuji et al, 1982; van der Wielen et al, 2023; Schneider and Doetsch, 1977). Since the low temperature prevented the movement of phages along pili, we propose that JBD30 particles are brought to the cell surface by pilus retraction.

## Tape measure protein and its role in JBD30 genome delivery

The interior of most of the tail tube is occupied by a trimer of tape measure proteins (gp46) (Figs. 1B and 5A). However, the cryo-EM density in the tail channel is not well resolved, except for the part in the baseplate core (Fig. 5A). The resolved density enabled building of the structure of the 33 C-terminal residues of the tape measure protein, which form a coiled coil (Fig. 5E; Appendix Fig. S8). Hydrophobic residues are buried inside the helical bundle, as in the tape measure protein of phage T5 (Linares et al, 2023). Residues Lys1149, Arg1151, and Gln1152 interact with Ser59, Asp61, Ser64, and Gln66 from domain I of the upper baseplate protein (Appendix Fig. S8D), and thus position the C-terminus of the tape measure protein in the middle of the tube. Methionines 1130 and 1134 from the C-terminal domain of the three tape measure proteins coordinate a putative ion positioned on the tail axis (Fig. 5E).

C-terminal domains of tape measure proteins from phages T5 (PDB-7zqb), 80α (PDB-6v8i), and JBD30 can be superimposed with an RMSD of C-alpha atoms lower than 0.353 Å for 70% of the residues available for the comparison (Appendix Fig. S8). The fold of the tape measure protein C-terminus is conserved among siphophages infecting gram-negative (JBD30: *P. aeruginosa*, T5: *Escherichia coli*) as well as gram-positive hosts (80α: *Staphylococcus aureus*) (Kizziah et al, 2020; Linares et al, 2023).

Using AfphaFold2 multimer (Evans et al, 2021) prediction and comparison with tape measure proteins of other phages, we attempted to estimate the role of the protein in infection. We propose that when the baseplate tip opens, the C-terminal domain of tape measure protein (residues 1128–1158) is released, followed by residues 1067–1127. At position 1066 of the JBD30 tape measure protein, there is a cleavage site for Arg-C proteinase (predicted with EXpasy Peptide Cutter). Unlike in phage T5, where there is a Zinc carboxypeptidase in this part of the tape measure protein, we were unable to identify a peptidase motif in the JBD30 sequence (Appendix Fig. S8B) (Boulanger et al, 2008). Residues 1022–1066 are predicted to fold into a hook composed of two α-helices and a short β-hairpin (Appendix Fig. S8C). A similar hook structure holds the tape measure protein of the T5 phage inside the baseplate tube after the expulsion of its C-terminal domain (Linares et al, 2023). We speculate that the JBD30 tape measure protein is proteolytically cleaved at position 1066, the C-terminal domain is released and the new C-terminus (residues 1022–1066) anchors the tape measure protein to the interior of the baseplate. Residues 930–1021 of the JBD30 tape measure protein are predicted to be unstructured (Appendix Fig. S8C). This unstructured chain is long enough to stretch over the length of the baseplate channel to the tip opening (Fig. S8C). The rest of the tape measure protein (residues 57–929) was predicted to form a transmembrane channel composed of α-helical segments (Fig. S8C) (Mistry et al, 2021). We speculate that the putative transmembrane channel serves for dsDNA translocation across the outer and inner *P. aeruginosa* membranes to enable genome delivery into the cytoplasm.

## Structure of procapsid

The heads of tailed phages assemble as procapsids (Aksyuk and Rossmann, 2011). The sample of JBD30 used for the collection of electron micrographs contained a fraction of procapsids, which enabled their structure determination (Appendix Fig. S1, Appendix Table S2). The procapsid of JBD30 has thicker walls and is more spherical than the mature capsid (Fig. 6A–C). The hexamers, built from the major capsid proteins, in the procapsid are compressed and skewed, as was observed in the procapsids of other phages (Chen et al, 2011; Bayfield et al, 2019; Huet et al, 2019; Fang et al, 2022). The diameter of the procapsid is 556 Å, which is 6.8% smaller than the diameter of the mature capsid (596 Å) (Figs. 2A and 6A). Pores along the central axes of the pentamers (15 Å diameter) and hexamers (13 Å diameter) are wide enough to enable the escape of scaffolding proteins from the procapsid (Fig. 6A,B). No decoration proteins are present at the procapsid surface, since the binding sites for the minor capsid proteins are not formed because the arrangement of major capsid proteins differs from that of the mature head (Fig. 6A). The expansion of the procapsid into a mature capsid is enabled by conformational changes to the major capsid protein. During capsid maturation the N-terminal arm (residues 6–25), which is an α-helix in the procapsid, extends, unfolds, and rotates ~40° anti-clockwise (Fig. 6D). The peripheral domain bends towards the capsomer centre and the extended loop straightens and becomes narrower (Fig. 6D).

The portal in the procapsid has twelvefold symmetry and is organized into clip, stem, wing, and crown domains, as in the virion (Fig. 6E). While the procapsid portal crown, wing, and stem domains are hidden inside the procapsid, the clip domain tightly interacts with the procapsid shell (Fig. 6E). During procapsid maturation, the portal complex elongates along its central axis, similarly as in phage P23-45 (Bayfield et al, 2019). The crown and wing domains remain inside the capsid, the latter rests on the inner capsid wall (Fig. 6E). Unlike in the procapsid, in the JBD30 virion, the stem domain helices α9 and α11 cross the capsid shell (Fig. 6E). The clip domain reaches further out of the capsid to enable the attachment of the adaptor complex (Figs. 3A and 6E).

## Cryo-electron tomography and fluorescence microscopy of JBD30 infection

Tomograms and cryo-TEM images of infected *P. aeruginosa* cells confirmed that bacteriophage JBD30 utilizes a type IV pilus for initial attachment (Fig. 7A). Two to ten minutes post-infection, phages attached along type IV pili using their tail fibres, however, some of them were already bound to cell poles (Fig. 7A). The particles attached to the cell surface were at various stages of genome release (Fig. 7A,B). *P. aeruginosa* cells produced outer membrane vesicles, possibly as a form of defence, since many phages attached to these vesicles (Augustyniak et al, 2022). After binding to the cell wall, the receptor binding protein legs slide aside and make room for the baseplate tip to touch the outer *P. aeruginosa* membrane (Fig. 7C). To deliver its genome into the cell cytoplasm, JBD30 forms a channel spanning the two membranes and the peptidoglycan layer covering *P. aeruginosa* cells (Fig. 7C). As discussed above, the channel is formed by the tail tape measure proteins ejected from the tail channel. The steps of the JBD30 replication cycle immediately following the genome delivery cannot be directly observed using cryo-EM until the stage of progeny particle assembly. Newly assembled procapsids were visible inside the infected cells 60 min post-infection (Fig. 7D; Appendix Fig. S9). Most of the cells in samples prepared 90 min post-infection were lysed (Fig. 7E; Appendix Fig. S9). The lysis of JBD30-infected cells occurs on average 85 min post-infection (Appendix Fig. S9).

Not all the cells from the exponentially growing culture of *P. aeruginosa* are equally sensitive to JBD30 infection, because of the differences in the numbers of pili formed by individual cells. Even when an excess of phage particles was added and some cells were infected by as many as 45 phage particles, there was still 6% of cells with no phage particles attached (Appendix Fig. S10). In control *P. aeruginosa* cells, 45% formed no pilus, 45% were pilated at one pole, and 10% had pili at both poles (Appendix Fig. S10). The predominantly unipolar location of type IV pili is determined by the location of extension and retraction ATPases pilT, pilB, and pilU (Burrows, 2012). While pilB and pilT are found at both cell poles, the distribution of pilU is unipolar (Chiang et al, 2005). Three minutes post JBD30 infection, *P. aeruginosa* cells formed the same number of pili as control cells (Appendix Fig. S10). It has been shown that the production of type IV pili by individual *P. aeruginosa* cell ranges from 0 to 35 pili per minute, and 80% of cells

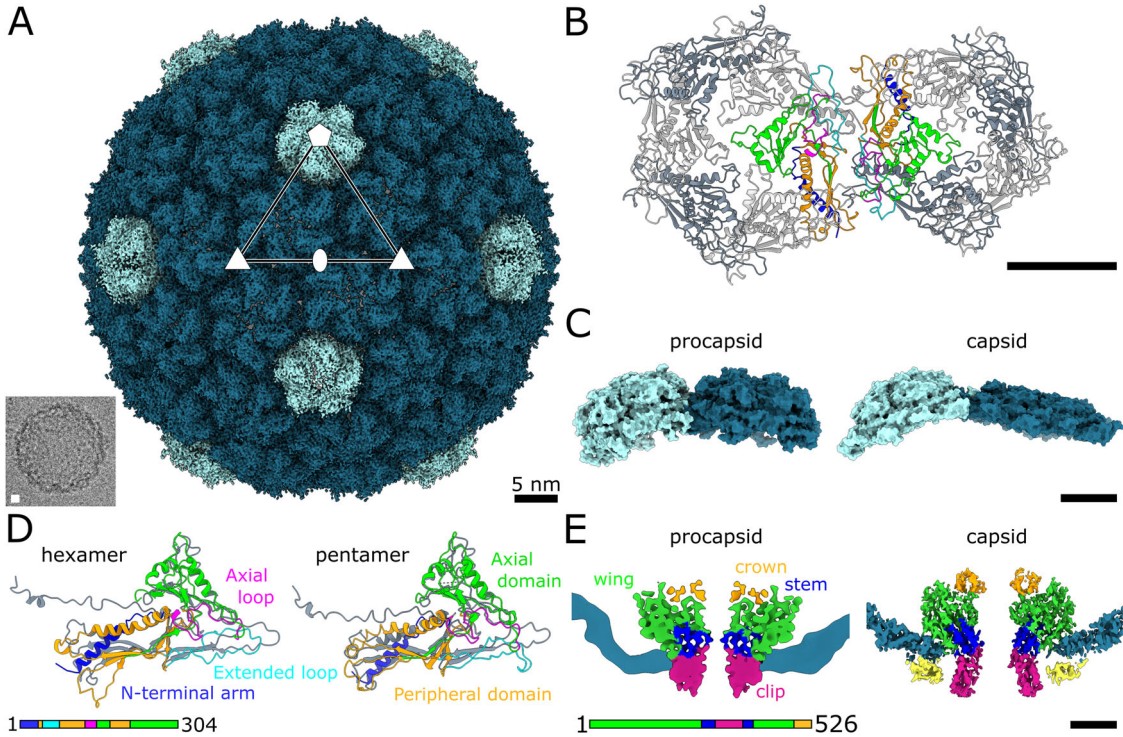

**Figure 6. Structure of JBD30 procapsid.**

(A) Surface representation of cryo-EM reconstruction of JBD30 procapsid. Pentamers of major capsid protein are highlighted in light blue, hexamers in turquoise. Positions of selected icosahedral five-, three-, and twofold symmetry axes are indicated. The inset shows an electron micrograph of a JBD30 procapsid. (B) Cartoon representation of major capsid proteins forming pentamer and hexamer of procapsid. Two major capsid proteins (one from the hexamer and one from the pentamer) are coloured according to domains as in panel (D). (C) Comparison of pentamers and hexamers from procapsid and capsid. Sideview of the major capsid protein pentamer and hexamer shown as molecular surfaces. (D) Comparison of major capsid proteins from procapsid and capsid. The major capsid proteins from the procapsid are coloured according to domains. Superimposed major capsid proteins from the capsid are shown in grey. Major differences occur in the position of the N-terminal arm and in the width of the extended loop and peripheral domain. (E) Comparison of interactions of portal complex with capsid proteins and in JBD30 procapsid and capsid. Left: Central slice through composite map of portal complex and procapsid. Right: Central slice through asymmetric reconstruction of capsid–portal interface. Density occupied by the major capsid proteins is shown in blue, density of minor capsid proteins is in yellow, density of the portal protein is coloured according to domain. Scale bars 5 nm.

were shown to form at least one pilus per 30 s (Koch et al, 2021). The infection of JBD30 depends on the initial attachment of the phage to the type IV pilus (Fig. 7) (Harvey et al, 2017). To image JBD30 cell attachment, the cells were plunge frozen at three min post-infection, which enabled the cells to undergo several cycles of pilus extension and retraction and gave phages ample opportunities to bind to them. Three minutes post-infection, 94% of cells had phage particles attached (Appendix Fig. S10).

The lattice SIM fluorescent microscopy of *P. aeruginosa* cells infected with DAPI-labelled JBD30 confirmed the crucial role of type IV pili in host cell recognition, and enabled the visualization of the JBD30 replication cycle (Fig. 7F–J, Appendix Figs. S11 and S12). Shortly after infection, the signal from DAPI-labelled phage DNA accumulated at bacterial cell poles (Fig. 7G, Appendix Figs. S11 and S13). This signal localization corresponds to the position of type IV pili growing from the cell poles (Burrows, 2012; Chiang et al, 2005). As the labelled phage genome was delivered into the cytoplasm, the signal from DAPI-labelled DNA moved inside the cells (Fig. 7H; Appendix Figs. S11 and S13). During later stages of infection, the DNA signal became more granular as the labelled DNA was incorporated into the procapsids of the newly assembled phage particles (Fig. 7I; Appendix Figs. S11 and S13). Finally, as the cells

lysed, we observed the release of phage progeny (Fig. 7J; Appendix Figs. S11 and S13).

## JBD30 replication cycle

The combination of data from cryo-electron tomography and fluorescent microscopy enabled us to propose a working model of the replication cycle of JBD30 (Fig. 8A–H). Bacteriophage JBD30 binds to a *P. aeruginosa* type IV pilus using its tail fibres (Figs. 5F, 7A, and 8B). This is in agreement with previously published data showing that glycosylation of type IV pili blocks phage infection (Harvey et al, 2017). After the retraction of pilus, the phage reaches the cell surface and attaches to the cell surface using the receptor binding protein tripod (Fig. 8C). The tripod legs then slide aside, which triggers the opening of the baseplate tip gated by three α-helices of baseplate hub trimer (Figs. 7B and 8D). The baseplate tip opens, the C-terminal domain of tape measure protein (residues 1067–1158) is extruded from the baseplate, and the protein is cleaved at residue 1066 (Linares et al, 2023). We hypothesize that the new C-terminus of the tape measure protein binds to the baseplate core (Appendix Fig. S8). The rest of the tape measure protein, containing 78% amino acids with hydrophobic or neutral

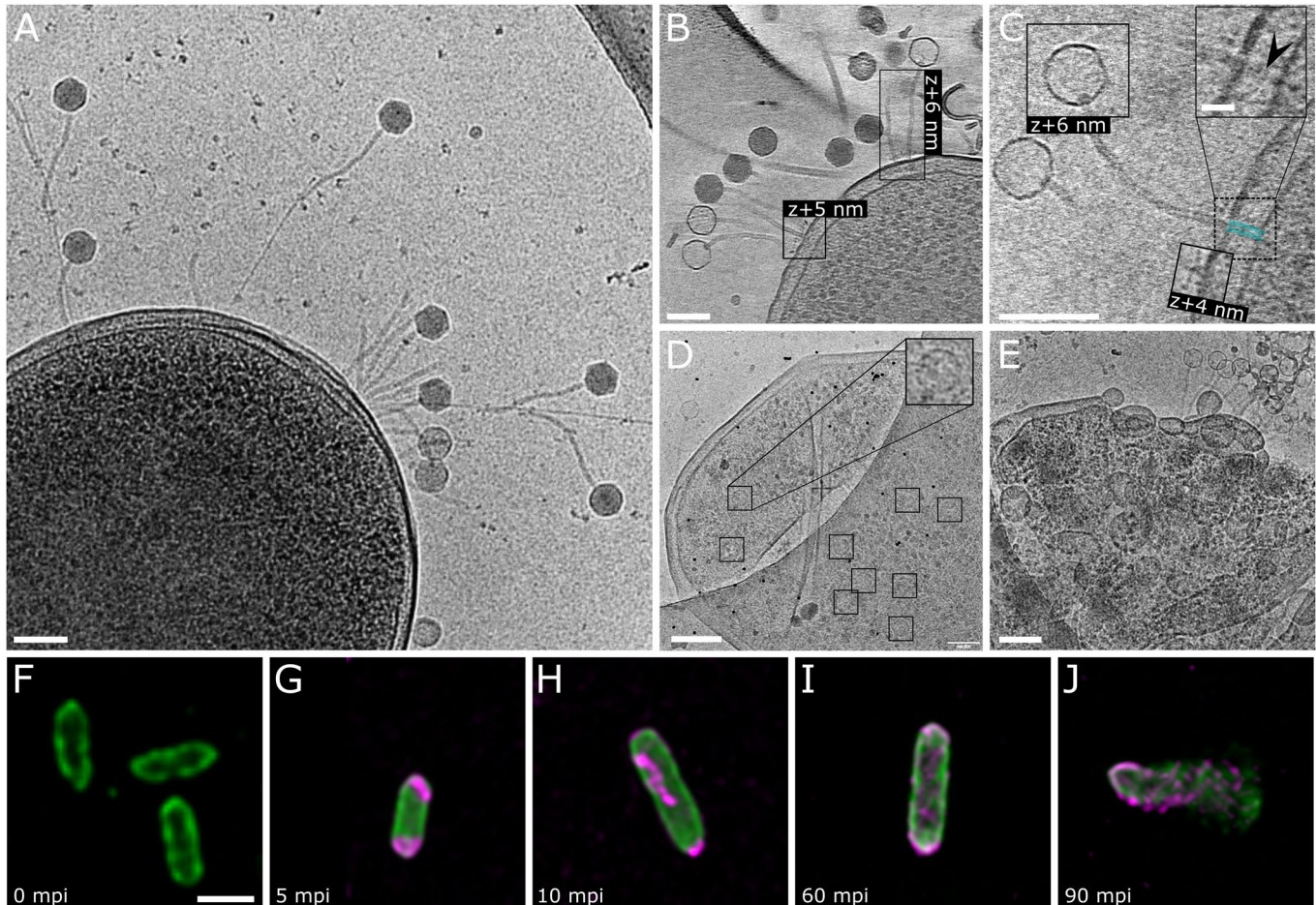

**Figure 7.  Replication cycle of phage JBD30 visualised using cryo-EM, cryo-ET, and lattice SIM fluorescence microscopy.**

(A) Electron micrograph showing binding of JBD30 virions to type IV pili. (B) Attachment of JBD30 to cell surface. The image shows a projection of a 12-nm-thick section from a tomogram of a cell at 30 min post-infection. (C) Ejection of JBD30 genome into the *P. aeruginosa* cell. Putative channel for DNA translocation spanning bacterial cell wall is highlighted by semi-transparent cyan lines. The scale bar in the inset represents 25 nm. (D) Assembly of phage progeny in infected cell 60 min post-infection. The inset shows detail of the empty procapsid. (E) Cell lysis at 90 min post-infection. Scale bars in (A–E) represent 100 nm. (F–J) Visualization of phage JBD30 replication in *P. aeruginosa* using fluorescence microscopy. The phage genome is labelled with DAPI (magenta), *P. aeruginosa* PAO1 *pilA-A86C* cells are stained with Alexa Fluor 488 C5 maleimide (green). Scale bar represents 1 μm, mpi—min post-infection. Images from individual channels are shown in Appendix Fig. S11. Please note that images shown in panels (F), (H), (L), (J) are identical to those displayed in Appendix Fig. S11A–E. (F) Non-infected cells of *P. aeruginosa*. (G) *P. aeruginosa* infected by JBD30. The phage signal is localized to the cell poles, since the phages utilize type IV pili for the initial cell binding. (H) Delivery of phage genome into host cell. (I) Assembly of new virions. The DAPI stain from the infecting phage particles remained in the cell and in the late stages of infection re-distributed to the newly synthetized DNA. (J) Cell lysis and release of phage progeny.

sidechains, is expelled from the tail tube, inserts into the membrane, and re-folds into a new conformation to form a channel for genome delivery into the host cytoplasm (Figs. 7C and 8E; Appendix Fig. S8) (Boulanger et al, 2008; Mahony et al, 2016). The empty JBD30 particle remains associated with the cell wall (Figs. 7D,E and 8F). JBD30 then employs anti-CRISPR protein AcrF1 to evade the host immune system (Bondy-Denomy et al, 2013) and quorum-sensing anti-activator protein Aqs1 to attenuate cell-to-cell communication, which would result in reduced pilus production (Appendix Table S1) (Shah et al, 2021). Based on the lysis-lysogeny decision controlled by the *gp1*, λ-like repressor cI, JBD30 DNA either integrates into the host chromosome (using *gp6* encoded transposase) or enters the lytic cycle to produce new virions (Fig. 8F,G; Appendix Table S1). The latent period lasts 85 min (Appendix Fig. S9), during which the new virions assemble

(Fig. 8G). The formation of the capsid and tail of siphophages occurs in separate pathways (Aksyuk and Rossmann, 2011). The JBD30 capsid assembles as a procapsid by the copolymerization of scaffolding proteins *gp33* and major capsid proteins *gp38* from the nucleation centre formed by the portal complex (Figs. 6, 7, and 8G; Appendix Table S1). The scaffolding protein is then cleaved by prohead protease *gp37* and exits the procapsid, probably through the pores in the centre of the capsomers (Fig. 6A,B). During procapsid maturation, major capsid proteins and portal proteins undergo a series of structural changes leading to capsid expansion. Portal protein stem and clip domains protrude from the capsid to enable the binding of head completion proteins (Fig. 6E). Trimers of minor capsid proteins bind to threefold and quasi-threefold axes of the mature capsids (Fig. 2A). These trimers increase capsid stability; however, they are not essential for phage viability

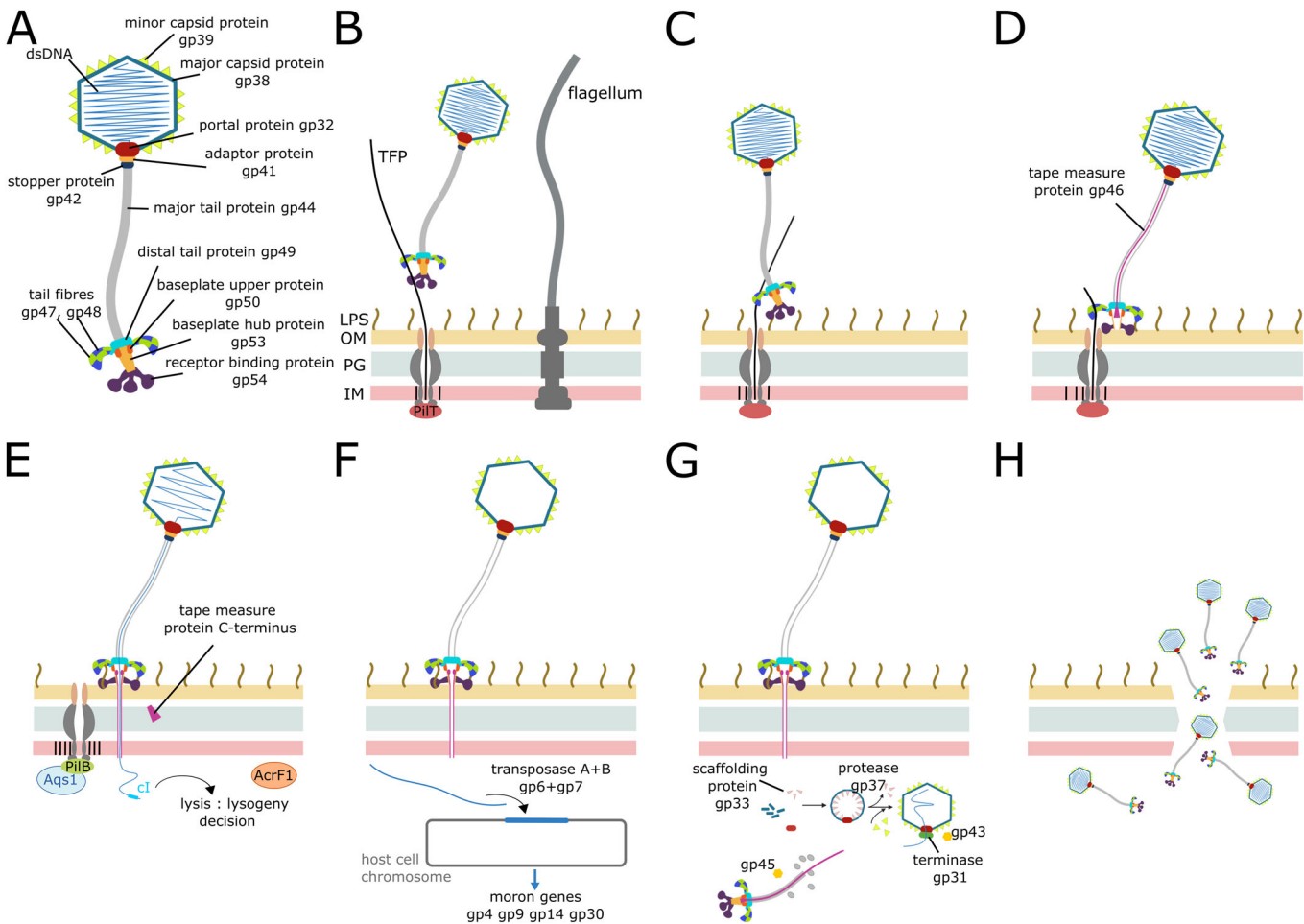

**Figure 8. Structure and proposed model of replication cycle of JBD30.**

(A) Scheme of JBD30 virion. (B–H) JBD30 replication cycle. (B) JBD30 virion attaches to the type IV pilus using its tail fibre. LPS—lipopolysaccharide, OM—outer membrane, PG—peptidoglycan layer, IM—inner membrane, TFP—type IV pilus, PilT—retraction ATPase. (C) As the pilus retracts, the phage particle is brought to the cell surface and binds to it using receptor-binding proteins. (D) Receptor-binding protein legs slide aside, and baseplate tip opens to release tape measure proteins. (E) Tape measure proteins form a channel for genome translocation into host cell. Based on the lysis-lysogeny decision, JBD30 DNA either integrates into the host chromosome (F) or enters the lytic cycle to produce new phage virions (G). PilB—assembly ATPase, Aqs1—quorum-sensing anti-activator, AcrF1—anti-CRISPR protein. (G) Capsids and tails of new phages are assembled in separate pathways and joined together. *gp43*—DNA packaging chaperone, *gp45*—tail assembly chaperone. (H) Replication cycle ends with lysis of host cell and phage progeny release. Created with BioRender.com.

(Qin et al, 2010). DNA is filled into the capsid in a headful packaging mechanism by the terminase encoded by *gp31* (George and Bukhari, 1981). After the DNA packaging is finished, the terminase is replaced by two rings of head completion proteins: an adaptor protein dodecamer and stopper protein hexamer (Fig. 3A). The latter forms an interface for tail binding (Fig. 4A). The assembly of the tail starts from the preassembled baseplate complex. The tape measure protein C-terminal domain is anchored in the baseplate core (Fig. 5A), and the tail tube is formed by 270 copies of major tail protein polymerizing around the tape measure protein into hexameric discs (Fig. 4A). Completed tails are joined with capsids. The JBD30 replication cycle ends with the host cell lysis and release of phage progeny. The first virions occur at 80 min post-infection, and the titer peaks at 100 min post-infection (Fig. 8H; Appendix Fig. S9). The combination of cryo-electron tomography and fluorescent microscopy has enabled

characterization of the replication cycle of phage JBD30, shedding light on its intricate mechanisms of host cell recognition, genome delivery, and progeny particle formation.

## Methods

### Phage propagation

Bacteriophage JBD30 was kindly provided by prof. Alan Davidson from the University of Toronto. *Pseudomonas aeruginosa* strain BAA-28 was used for JBD30 propagation. Overnight culture of *P. aeruginosa* BAA-28 was diluted into fresh LB medium (1:1000) and grown to OD$_{600}$ ≈ 0.3 (~10$^8$ CFU/ml) at 37 °C, 250 RPM. Phage lysate was added to the culture (MOI 1), and the mixture was incubated for 1 h at 30 °C and 60 RPM, 1 h at 30 °C and 40 RPM,

1 h at RT, and stored overnight at 4 °C. The cell debris were removed by centrifugation at $5000 \times g$ at 4 °C for 20 min, and the supernatant filtered through a 0.2 µm filter (SFCA membrane, Corning). The titer of phage lysate was determined using a plaque assay (Kropinski, 2018).

## Phage purification

The phage lysate (500 ml) was centrifuged in a 50.2 Ti rotor (Beckman Coulter) at $54,000 \times g$ and 10 °C for 2.5 h (Optima XPN-80 Ultracentrifuge, Beckman Coulter). The phage pellet was resuspended in 200 µl of phage buffer (10 mM MgSO$_4$, 10 mM NaCl, 50 mM Tris pH 8) at 4 °C O/N. The whole volume of the sample was pipetted onto the top of a CsCl step density gradient (3 ml of 1.45 g/ml CsCl, 3 ml of 1.50 g/ml CsCl and 3 ml of 1.70 g/ml CsCl in phage buffer) and centrifuged for 7.5 h at 35,000 RPM, 10 °C (Ultracentrifuge Beckman Coulter, SW 41Ti rotor, 13.2 ml Ultra Clear tubes). Bands corresponding to concentrated phage particles were collected using a syringe with an 0.8 mm gauge needle. CsCl was removed by dialysis against an excess of phage buffer at 4 °C for 2 days. Cellulose dialysis tubing type 27/32" with 14 kDa MWCO (CarlRoth) was used.

## One-step growth curve

Phage growth parameters were determined using the One-step growth protocol (Kropinski, 2018). Briefly, 10 ml of bacterial culture of $OD_{600} \approx 0.3$ (~$10^8$ CFU/ml) was mixed with 100 µl of phage lysate ($10^8$ PFU/ml). After 10 min of incubation at 37 °C and 100 RPM to allow the adsorption of phages to bacterial cells, the sample was centrifuged ($14,000 \times g$, 90 s, RT) and resuspended in 1 ml of LB medium, then centrifuged again and resuspended in 1 ml of LB medium. Supernatants were saved for assessing the amount of non-adsorbed phages. A volume of 10 µl (from the total 1 ml of the cell suspension) was transferred into a pre-warmed 10 ml of LB medium and incubated at 37 °C and 80 RPM. Samples for plaque assays were taken regularly over the course of the experiment.

## Identification of structural proteins

A 10 µl volume of purified bacteriophage JBD30 sample ($10^7$ PFU/ml) was mixed with 2.5 µl of 5x concentrated Protein loading buffer, heated to 95 °C for 3 min, and then loaded into the wells in 15% polyacrylamide gel. The electrophoresis was run in Tris-glycin buffer containing sodium dodecyl sulfate (SDS) at 180 V until the samples became concentrated next to the stacking gel and then with a constant voltage of 250 V for ~30 min. The protein bands were visualised using Instant Blue™ Protein Stain. Protein bands were cut out from the gel, digested using trypsin at 40 °C for 2 h, and analysed by MALDI MS/ MS. The software FlexAnalysis 3.4 and MS BioTools (Bruker Daltonics) were used for data processing and analysis. Sequences in exported MS/MS spectra were searched against the National Center for Biotechnology Information database and a local database supplied with the expected protein sequences using the software Mascot (Matrix Science, London, UK). The mass tolerance of peptides and MS/MS fragments for MS/MS ion searches were 50 parts per million and 0.5 Da, respectively. The oxidation of methionine and propionyl-amidation of cysteine were set as optional modifications for all searches, as well as one enzyme mis cleavage. Only peptides with a statistically significant peptide score ($P < 0.05$) were considered.

## Preparation of samples for fluorescence microscopy

Phage lysate (2 ml) was treated with Turbonuclease for 1 h at 37 °C, to ensure that only encapsidated DNA remained in the sample. The labelling of the phage genome with DAPI (Roche, 5 mg/ml) in ratio 1:1000 (v:v) was performed overnight at 4 °C. Labelled phage particles were pelleted by ultracentrifugation ($54,000 \times g$, 2.5 h), resuspended in 2 ml of phage buffer and pelleted again. The final pellet was resuspended in 200 µl of phage buffer. Bacterial strain *P. aeruginosa* PAO1 pilA-A86C was kindly donated by Yves V. Brun (Ellison et al, 2019). Overnight culture of *P. aeruginosa* PAO1 *pilA-A86C* was used to inoculate 5 ml of LB medium (1:1000) and incubated at 37 °C, 250 RPM until $OD_{600} \approx 0.3$ ($10^8$ CFU/ml) was reached. Cells from 1 ml of the culture were harvested by centrifugation (1 min, $8000 \times g$) and resuspended in 500 µl of PBS buffer. This washing step was repeated twice. Labelling with Alexa Fluor™ 488 C$_5$ Maleimide (Thermo Fisher) was performed in a 1:1000 ratio (v:v) at 37 °C for 1 h in the dark. To remove the unbound fluorescent dye, the labelled cells were centrifuged (1 min, $8000 \times g$) and resuspended in fresh 500 µl of PBS. *P. aeruginosa* PAO1 pilA-A86C labelled with Alexa Fluor™ 488 (50 µl) was pipetted into the wells of a glass-bottomed cell culture plate (Greiner Bio-One CELLview™) and DAPI pre-stained phages were added at MOI 100. The plate was incubated at 37 °C and 80 RPM. To stop the infection at the selected times, 50 µl of fixation solution (8% w/v formaldehyde, 0.2% w/v glutaraldehyde, 50 mM phosphate buffer pH 7.5) was added to the wells.

## Fluorescence microscopy data acquisition and processing

Fixed samples of phage-infected cells were imaged using a ZEISS Elyra 7 Lattice SIM$^2$ microscope equipped with $2 \times$ PCO edge sCMOS cameras using a Plan-Apochromat 63x/1.46 oil objective. 405 nm and 488 nm lasers were used for the excitation of DAPI and Alexa Fluor™ 488, respectively. Z-stacks of images acquired in sequential mode were processed using the Zen Black SIM$^2$ algorithm and projected as maximum intensity projections.

## Preparation of samples for cryo-TEM and cryo-ET

A sample with purified JBD30 (4 µl of $10^{11}$ PFU/ml) was applied onto a Quantioil™ grid (2/1, Cu, mesh 300) glow-discharged in H/O plasma using a Gatan Solarus II. Grids were blotted (blotting force 0, blotting time 2 s, 100% humidity, waiting time 15 s), plunge-frozen in liquid ethane using a Vitrobot Mark IV, and stored in liquid nitrogen. *P. aeruginosa* cells for cryo-electron tomography sample preparation were grown in LB medium at 37 °C, 250 RPM to $OD_{600} \approx 0.3$ ($10^8$ CFU/ml). At time 0, the phage lysate was added to the culture at a multiplicity of infection of 500. The infected culture was incubated at 30 °C, 60 RPM. Two minutes post infection, the cells were pelleted (1 min, $5000 \times g$) to increase the cell density and remove the non-adsorbed phages. The pellet was resuspended in fresh LB medium to reach $OD_{600} \approx 9$. Samples for vitrification were taken at distinct time points over the course of infection. A volume of 4 µl of the infected cells was applied onto a Quantifoil™ grid (2/1, Cu, mesh 300) glow-discharged in H/O plasma using a Gatan Solarus II. Before sample application, 4 µl of gold fiducials (BSA Gold Tracer 10 nm, AURION) were applied onto the grid, and after 30 s manually blotted with a piece of filter paper. Grids with applied cells were blotted (blotting force 0, blotting time 2 s, 100%

humidity, wait time 15 s), plunge-frozen in liquid ethane using a Vitrobot Mark IV, and stored in liquid nitrogen.

## Determination of the effect of temperature on the speed of translocation of pilus-attached JBD30 to the cell surface

*P. aeruginosa* cells were grown in LB medium at 37 °C, 250 RPM to $OD_{600} \approx 0.3$ ($10^8$ CFU/ml) and incubated 1 h on ice to slow down their metabolic activity. Control cells were vitrified on grids for electron microscopy using the following procedure: The cells were pelleted (1 min, $5000 \times g$, 4 °C). The pellet was resuspended in fresh ice-cold LB medium to reach $OD_{600} \approx 9$. A volume of 4 µl of the cell suspension was applied onto a Quantifoil™ grid (2/1, Cu, mesh 300) and plunge-frozen in liquid ethane using a Vitrobot Mark IV (blotting force 0, blotting time 2 s, wait time 10 s, 100% humidity, 4 °C), and stored in liquid nitrogen. JBD30 lysate was pre-cooled on ice and added to a *P. aeruginosa* culture at a multiplicity of infection of 500. The infected culture was incubated on ice for 15 min. A fraction of the cells was vitrified for cryo-EM. The remaining cells were split in half. The first group was incubated for an additional 15 min on ice and vitrified for cryo-EM. The other group was incubated at 37 °C and 60 RPM for 15 min and vitrified for cryo-EM. One hundred cells from each group were imaged using cryo-TEM, and the number of cells with pili and phages attached to pili or a cell surface were determined.

## Cryo-EM data acquisition and initial processing

Dataset 1 (11,300 micrographs) for single-particle analysis was collected using a Titan Krios electron microscope operating at 300 kV, equipped with a K2 Summit detector operating in counting mode. The detector was behind an energy filter operating in zero-loss mode with a 20 e⁻V slit inserted. NanoProbe mode was used to ensure parallel illumination. Coma-free alignments were performed to remove the residual beam tilt. A total dose of 30 e⁻Å⁻² was used for imaging under low-dose conditions. The 5-s exposure was fractionated into 25 frames and saved as a movie. The defocus applied during data acquisition varied from −0.6 µm to −1.6 µm. A nominal magnification of 130,000× resulted in a calibrated pixel size of 1.04 Å. Dataset 2 (12,356 micrographs) was collected using the Titan Krios operated at 300 kV, equipped with a K3 detector operating in counting mode. The detector was behind an energy filter operating in zero-loss mode with a 10 e⁻V slit inserted. A total dose of 34 e⁻Å⁻² was used for imaging under low-dose conditions. The 2-s exposure was split into 40 frames and saved as a movie. The defocus applied during data acquisition varied from −0.6 µm to −1.6 µm. A nominal magnification of 105,000× resulted in a calibrated pixel size of 0.8336 Å. Dataset 3 (13,769 micrographs) was collected using the Titan Krios operated at 300 kV, equipped with a K2 detector operating in counting mode. The detector was behind an energy filter operating in zero-loss mode with a 20 e⁻V slit inserted. The 7.5-s exposure was split into 40 frames and saved as a movie. The defocus applied during data acquisition varied from −0.6 µm to −1.6 µm. A nominal magnification of 130,000× resulted in a calibrated pixel size of 1.057 Å. A total dose of 53 e⁻Å⁻² was used for imaging under low-dose conditions. Automated data acquisition was performed using the EPU software. Dataset 4 (3568 micrographs) for single-particle analysis

of a JBD30 baseplate attached to a *P. aeruginosa* type IV pilus was collected using the Titan Krios operated at 300 kV, equipped with a K3 detector operating in counting mode. Data acquisition was performed semi-automatically using SerialEM (Mastronarde, 2005). First, low magnification images of the whole grid were acquired to choose the grid squares with optimal ice thickness and cell coverage. Selected grid squares were then acquired as a polygon montage at a nominal magnification of 4800× and defocus of −80 µm. The detector was behind an energy filter operating in zero-loss mode with an inserted 50 e⁻V slit. An exposure of 0.5 s was used. The areas for data acquisition were manually selected on bacterial cell poles positioned in the holes of the holey carbon coating film on the grid. The acquisition of selected areas was done using the "Multishot" acquisition setup in SerialEM (Mastronarde, 2005) at a nominal magnification of ×42,000, resulting in a calibrated pixel size of 2.2 Å. The defocus applied during data acquisition varied from −1.6 µm to −2.6 µm. The 4-s exposure was saved as a 40-frame movie. A total dose of 22.4 e⁻Å⁻² was used for imaging under low-dose conditions. The detector was behind an energy filter operating in zero-loss mode with a 10 e⁻V slit inserted. MotionCor2 (Zheng et al, 2017) was used for the global and local motion correction of acquired movies, which were subsequently saved as dose-weighted micrographs. Defocus values were estimated from aligned non-dose weighted micrographs using the programme Gctf 1.06 (Zhang, 2016).

## Cryo-ET data acquisition and processing

Prepared grids were used for tomography data acquisition using a Titan Krios microscope equipped with a K2 detector behind an energy filter with a 20 e⁻V slit inserted. Tomography tilt series were recorded in the angular range of −60° to +60° with a 2° increment in a dose-symmetric bidirectional tilt-scheme in low dose mode using SerialEM (Hagen et al, 2017). The nominal defocus value was −6 µm. Each image was acquired as a movie of 3 frames. A dose of 1.8 e⁻Å⁻² per tilt was used, resulting in a total dose of 74 e⁻Å⁻² for each tilt series. Tomograms were collected at a magnification of 53,000×, resulting in a pixel size of 2.78 Å. The software eTomo (IMOD (Kremer et al, 1996)) was used for tilt-series alignment and tomogram reconstruction.

## Icosahedral reconstruction of JBD30 capsid

A total of 24,991 phage capsids were automatically picked from micrographs of dataset 1 using the programme crYOLO (Wagner et al, 2019). Particles were extracted (840 × 840 px, 1.04 Å/px) and subjected to two rounds of 2D classification, resulting in 12,700 particles. The initial model was prepared de novo using the Stochastic Gradient Descent algorithm (Punjani et al, 2017) in RELION 3.1 (Zivanov et al, 2018) using binned data (256 × 256 px, 3.4125 Å/px). The mask was created using the programmes UCSF Chimera (Pettersen et al, 2004) and relion_mask_create (Scheres, 2012). The initial 3D refinement with imposed icosahedral symmetry was followed by 3D classification into three classes omitting the orientation search and using orientations from the previous 3D refinement. Two classes resulted in highly resolved particles. Particles from the class of capsids decorated with minor capsid proteins (5360 particles) were selected for the subsequent rounds of 3D refinement and reextracted unbinned (840 × 840 px,

1.04 Å/px). Subsequent steps of 3D refinement, CTF refinement and reconstruction using Ewald sphere correction led to the final map which was threshold-masked, divided by the modulation transfer function and B-factor sharpened during post-processing in RELION 3.1 (Zivanov et al, 2018). Icosahedral reconstruction of the phage JBD30 capsid of the native virion and empty particle from dataset 2 was performed using the same strategy. The second round of 2D classification led to 5167 particles of virion capsids and 3037 particles of empty capsids. After 3D classification, the number of particles retained for further reconstructions was 4713 for the full particles and, 2666 for the empty.

### Fivefold symmetrized reconstruction of JBD30 capsid

Images of particles with orientations assigned from the final icosahedral 3D refinement were downscaled and re-extracted (256 × 256 px, 2.73525 Å/px) and aligned to have the connectors oriented along the z-axis using several rounds of 2D classification with masks (created in FIJI (Schindelin et al, 2012)) that only included the space for the connector. The orientation search was omitted during the classifications, and particle orientations and shifts were taken from the icosahedral 3D refinement. Because of the embedding in a thin layer of vitreous ice, the JBD30 particles were oriented with their tails pointing along the projection plane. Therefore, the classification resulted in classes of particles with the portal complexes oriented along one of the six directions of fivefold vertices from the icosahedral alignment deviating the least from the projection plane. Particles from each class were rotated to orient the portal complexes along the z-axis using RELION 3.1 (Zivanov et al, 2018) and custom-written script math_star.py (GitHub/fuzikt). The map from the icosahedral 3D refinement was rescaled (relion_image_handler) and used as an initial model for the first 3D refinement of the capsid in C5 symmetry, where only local searches of the Euler rot angle were allowed. A mask that included the capsid and portal was created using the programmes UCSF Chimera (Pettersen et al, 2004) and relion_mask_create (Scheres, 2012). Subsequent steps of 3D refinement, CTF refinement, and Ewald sphere correction (Zivanov et al, 2018) were done using unbinned data (840 × 840 px, 0.8336 Å/px). The final map was threshold-masked, divided by the modulation transfer function, and B-factor sharpened during post-processing.

### Asymmetric reconstruction of JBD30 capsid

Reextracted binned particles (256 × 256 px, 2.73525 Å/px) from the fivefold symmetrized capsid reconstruction were symmetry expanded to C1 using relion_particle_symmetry_expand (Scheres, 2012). Masked 3D classification without orientational alignment divided the particles into 5 classes. Unique particles were selected using the select_rand_sym_copy_ptcls.py script from starpy (GitHub/fuzikt) from the most populated class, and used for 3D refinement with only local searches. The final 3D refinement was performed on unbinned (840 × 840 px, 0.8336 Å/px) data. The final map was masked, and B-factor sharpened.

### Reconstruction of JBD30 connector complex

Fivefold symmetrized reconstruction of the capsid gave particles with connectors oriented along the z-axis. Using UCSF Chimera (Pettersen et al, 2004) and the map of fivefold symmetrized capsid reconstruction, the distance between the box centre and connector centre was determined. A modified script, localized_reconstruction.py (Ilca et al, 2015) (GitHub/fuzikt), was used to extract particles from micrographs centred on the connector (512 × 512 px, 0.8336 Å/px). The obtained particles were binned (128 × 128 px, 3.3344 Å/px) and reconstructed, applying C6 symmetry using relion_reconstruct, taking the orientations from the C5 reconstruction. This map served as an initial model for further refinements. The segmentation tool (Pintilie et al, 2010) in UCSF Chimera (Pettersen et al, 2004) and relion_mask_create (Scheres, 2012) were used for mask preparation. After initial 3D refinement in C6 symmetry, the particles were re-extracted (512 × 512 px, 0.8336 Å/px) and the initial model and mask were rescaled (using relion_image_handler (Scheres, 2012)). Initial 3D refinement in C6 symmetry was followed by 3D refinement with imposed twelvefold symmetry and 3D classification without particle orientation alignment. Particles from the class representing the best resolved features were used for subsequent rounds of 3D refinement and CTF refinement. The final map was threshold-masked, divided by the modulation transfer function, and B-factor sharpened during post-processing. This strategy was applied to resolve the connector structure of both the virion and empty particle.

### Reconstruction of JBD30 stopper protein

Particles from the C6 symmetrized refinement of the connector complex were used as an input for C6 symmetrized refinement of the stopper protein. Using the script localized_reconstruction.py (Ilca et al, 2015), the particles centred on the stopper protein were extracted from micrographs (512 × 512 px, 0.8336 Å/px). The map reconstructed using relion_reconstruct (Scheres, 2012) was used as an initial model. A mask covering only the stopper protein was prepared using a combination of UCSF Chimera (Pettersen et al, 2004) and relion_mask_create (Scheres, 2012). RELION 3.1 (Zivanov et al, 2018) 3D refinement with imposed C6 symmetry was performed with only local angular searches around previously determined orientations. Finally, the map was masked and B-factor sharpened.

### Asymmetric reconstruction of JBD30 connector

An asymmetric reconstruction of JBD30 the connector was calculated using the previously described strategy for P68 phage (Hrebík et al, 2019). Particles from the C12 symmetrized connector reconstruction were symmetry expanded to C1 using relion_particle_symmetry_expand (Scheres, 2012). A mask including the connector and its inner part was prepared using UCSF Chimera (Pettersen et al, 2004) and relion_mask_create (Scheres, 2012). The masked 3D classification without alignment and with regularization factor 33 separated the particles into 12 classes. The most populated class that exhibited interactions between the portal proteins and dsDNA inside the portal channel was chosen for further refinement steps. Selected unique particles were subjected to the final round of masked 3D refinement in C1 symmetry with only local searches. For the better visualization of contacts, B-factor blurring was applied during the post-processing step.

## Icosahedral reconstruction of JBD30 procapsid

Procapsids were manually boxed using cryolo_boxmanager.py (Wagner et al, 2019), extracted (720 × 720 px, 1.057 Å/px) and subjected to 2D classification. The initial model in icosahedral symmetry was prepared de novo using the Stochastic Gradient Descent (Punjani et al, 2017) algorithm implemented in Relion 3.1 (Zivanov et al, 2018) using binned particles (256 × 256 px, 2.972813 Å/px). The mask was created using the programmes UCSF Chimera (Pettersen et al, 2004) and relion_mask_create (Scheres, 2012). The initial 3D refinement with imposed icosahedral symmetry was followed by 3D classification into three classes without an orientational search using the particle orientations from the previous refinement. Particles from selected classes were unbinned (720 × 720 px, 1.057 Å/px) and used for subsequent steps of 3D refinement, CTF refinement and reconstruction with Ewald sphere correction applied using RELION 3.1 (Zivanov et al, 2018). The final icosahedrally symmetrized map was threshold-masked, divided by the modulation transfer function, and B-factor sharpened during post-processing.

## Fivefold symmetrized reconstruction of JBD30 procapsid

Procapsids were manually boxed using cryolo_boxmanager.py (Wagner et al, 2019), extracted (720 × 720 px, 1.057 Å/px), and subjected to 2D classification. Particles from selected classes were re-extracted (256 × 256 px, 2.972813 Å/px) and used for de novo C5 symmetrized initial model calculation using the Stochastic Gradient Descent algorithm (Punjani et al, 2017) implemented in RELION 3.1 (Zivanov et al, 2018). The mask was created using the programmes UCSF Chimera (Pettersen et al, 2004) and relion_mask_create.py (Scheres, 2012). The initial 3D refinement with imposed fivefold symmetry was followed by 3D classification into three classes without an alignment step. Particles from selected classes were unbinned (720 × 720 px, 1.057 Å/px) and used for another round of 3D refinement. The final C5 symmetrized map was threshold-masked, divided by the modulation transfer function, and B-factor sharpened during post-processing.

## Localized reconstruction of portal from JBD30 procapsid

Vertexes from the reconstruction of the icosahedral procapsid were extracted (256 × 256 px, 1.04 Å/px) using localized_reconstruction.py (Ilca et al, 2015) and subjected to 3D refinement in C12 symmetry. The C12 symmetrized map was used as an initial model for 3D classification of the extracted vertices. Particles were classified into 6 classes without the alignment step. Particles from the class exhibiting the portal features were retained. The script select_maxprob_sym_copy_ptcls.py (GitHub/fuzikt) was used to ensure that only one particle (representing one procapsid vertex) from each procapsid was present in the final particle set. The initial model for subsequent C12 symmetrized 3D refinement with only local angular searches was calculated using relion_reconstruct.py (Scheres, 2012). The mask was prepared using UCSF Chimera (Pettersen et al, 2004), Segger (Pintilie et al, 2010) and relion_mask_create (Scheres, 2012). Several rounds of 3D refinement and 3D classification led to the final C12 symmetrized map, which was threshold-masked, divided by the modulation transfer function, and B-factor sharpened during post-processing.

## C3 symmetrized reconstruction of JBD30 baseplate and baseplate core

CrYOLO (Wagner et al, 2019) was used for automatic particle picking of JBD30 baseplates from dataset 2. In total 8376 particles were extracted (540 × 540 px, 0.8336 Å/px) and subjected to 2D classification. Some of the 2D class averages of the baseplates exhibited threefold symmetry, which was subsequently imposed during the 3D reconstruction process. Particles from selected classes were re-extracted and binned (135 × 135 px, 3.30988 Å/px). The Stochastic Gradient Descent (Punjani et al, 2017) method in RELION 3.1 (Zivanov et al, 2018) was used for de novo initial model generation in C3 symmetry. After initial 3D refinement and 3D classification, 1780 particles representing the best-resolved features were reextracted without binning (540 × 540 px, 0.8336 Å/px) and used for subsequent steps of 3D and CTF refinement. The final map of the whole baseplate in C3 symmetry was threshold-masked, divided by the modulation transfer function, and B-factor sharpened during post-processing. A mask including only the central part of the baseplate (without tail fibres and without the receptor binding proteins) was created using the programmes UCSF Chimera (Pettersen et al, 2004) and relion_mask_create (Scheres, 2012). Two rounds of masked 3D refinement and CTF refinement led to the final C3 symmetrized map of the central part of the baseplate. This map was divided by the modulation transfer function and B-factor sharpened during post-processing.

## Asymmetric reconstruction of JBD30 baseplate

Particles (1780) from the C3 symmetrized reconstruction of the whole baseplate were subjected to single-class 3D classification with symmetry relaxation from C3 symmetry to C1 with a regularization factor of four. Subsequent 3D refinement in C1 symmetry with only local angular searches used a mask prepared using UCSF Chimera (Pettersen et al, 2004), Segger (Pintilie et al, 2010) and relion_mask_create (Scheres, 2012). The final map was divided by the modulation transfer function and B-factor sharpened during the post-processing step.

## Reconstruction of JBD30 tail

EMAN e2helixboxer.py (Tang et al, 2007) was used for JBD30 tail boxing. A refinement imposing helical symmetry on the tail segments composed of 3 to 5 discs was performed, however the flexibility and bending of the tail complicated the alignment. Better results were achieved using a reconstruction of the final tail discs above the baseplate. Tail particles (256 × 256 px, 0.8336 Å/px) were extracted using the script localized_reconstruction.py (Ilca et al, 2015). A C6-symmetrized initial model was calculated using relion_reconstruct (Scheres, 2012). The UCSF Chimera segmentation tool (Pintilie et al, 2010) and relion_mask_create (Scheres, 2012) were used for the preparation of a mask that included two tail discs. Initial 3D refinement in C6 symmetry was followed by 3D classification without alignment, resulting in 1759 particles. These particles were used for the final C6 symmetrized refinement using a mask including two tail discs. This map was used for the determination of helical rise and twist.

## Reconstruction of JBD30 receptor binding protein

The script localized_reconstruction.py (Ilca et al, 2015) was used to extract 5340 sub-particles (256 × 256 px, 0.8336 Å/px) centred on the receptor binding protein trimers from the final C3 symmetrized baseplate reconstruction. An initial model of the receptor binding protein trimer was calculated using relion_reconstruct (Scheres, 2012). UCSF Chimera (Pettersen et al, 2004), Segger (Pintilie et al, 2010) and relion_mask_create (Scheres, 2012) were used for mask preparation. Initial 3D refinement in C3 symmetry was followed by 3D classification into three classes without an alignment step, which resulted in 4088 particles. These particles were used for the final round of 3D refinement with imposed C3 symmetry. The final map was divided by the modulation transfer function and B-factor sharpened during post-processing.

## Reconstruction of JBD30 tail fibres

Particles from the final refinement of the C3 symmetrized baseplate reconstruction (540 × 540 px, 0.8336 Å/px) were used for the extraction of 5334 sub-particles (256 × 256 px, 0.8336 Å/px) centred on JBD30 tail fibres using the script localized_reconstruction.py (Ilca et al, 2015). An initial model for the tail fibre reconstruction was calculated using relion_reconstruct (Scheres, 2012). UCSF Chimera (Pettersen et al, 2004), Segger (Pintilie et al, 2010) and relion_mask_create (Scheres, 2012) were used for mask preparation. Initial 3D refinement in C1 symmetry was followed by 3D classification into three classes without an alignment step, resulting in 3927 particles. These particles were used for the final round of 3D refinement in C1. The final map was divided by the modulation transfer function and B-factor sharpened during post-processing.

## Reconstruction of JBD30 baseplate attached to type IV pilus

Aligned and motion corrected micrographs (3568) from dataset 4 were imported to cryoSPARC 4.0.0 (Punjani et al, 2017). Defocus values were estimated using Patch-based CTF estimation. After the inspection of curated micrographs, 1823 micrographs were accepted for further processing. The baseplates bound to a type IV pilus were manually picked (360 × 360 px, 2.2 Å/px) and subjected to 2D classification resulting in 603 particles. A composite map of JBD30 baseplate was low-pass filtered to 30 Å and used as an initial model for the first Non-uniform refinement (Punjani et al, 2020) with implemented global CTF and defocus refinement in C1 symmetry. A mask for subsequent rounds of Local refinement was prepared in UCSF Chimera (Pettersen et al, 2004), Segger (Pintilie et al, 2010) and relion_mask_create (Scheres, 2012). The final map was divided by the modulation transfer function and B-factor sharpened during post-processing in RELION 3.1 (Scheres, 2012).

## Model building and refinement

Maps from the refinement were cropped and normalized. Molecular models were built manually using the programme Coot version 0.9.8.7 (Emsley et al, 2010), and iteratively refined in real space using Phenix (Afonine et al, 2018) (version Phenix 1.20.1). After each iteration, the geometry of the model was evaluated using the MolProbity (Williams et al, 2018) tool, and geometry outliers were manually inspected and fixed in Coot (Emsley et al, 2010) (van Kempen et al, 2023; Kucukelbir et al, 2014; Paysan-Lafosse et al, 2023; Zimmermann et al, 2018).

# Data availability

Cryo-EM maps and structure coordinates were deposited into the Electron Microscopy Data Bank (EMDB) and Protein Data Bank (PDB), respectively, with the following accession numbers: (i) Structures of bacteriophage JBD30 virion: capsid computed with imposed icosahedral symmetry EMDB-19280 and PDB-8RKN, capsid computed with imposed fivefold symmetry EMDB-19274, asymmetric capsid reconstruction EMDB-19272, connector complex computed with imposed twelvefold symmetry EMDB-19267 and PDB-8RKB, stopper computed with imposed sixfold symmetry EMDB-19265 and PDB-8RK9, tail computed with imposed sixfold symmetry EMDB-19264 and PDB-8RK8, baseplate computed with imposed threefold symmetry EMDB-19263 and PDB-8RK7, baseplate core computed with imposed threefold symmetry EMDB-19262 and PDB-8RK6, asymmetric baseplate reconstruction EMDB-19261, asymmetric reconstruction of tail fibres EMDB-19260 and PDB-8RK5, receptor binding protein computed with imposed threefold symmetry EMDB-19259 and PDB-8RK4; (ii) Structures of bacteriophage JBD30 virion decorated with minor capsid protein trimers: capsid computed with imposed icosahedral symmetry EMDB-19270 and PDB-8RKC, capsid computed with imposed fivefold symmetry EMDB-19271, asymmetric capsid reconstruction EMDB-19269, asymmetric reconstruction of capsid–connector interface EMDB-19268; (iii) Structures of bacteriophage JBD30 empty particle: capsid computed with imposed icosahedral symmetry EMDB-19281 and PDB-8RKO, capsid computed with imposed fivefold symmetry EMDB-19273, connector complex computed with imposed twelvefold symmetry EMDB-19266 and PDB-8RKA; (iv) Structures of bacteriophage JBD30 procapsid: procapsid computed with imposed icosahedral symmetry EMDB-19285 and PDB-8RKX, procapsid computed with imposed fivefold symmetry EMDB-19258, procapsid portal computed with imposed twelvefold symmetry EMDB-19257; (v) Composite map of bacteriophage JBD30 baseplate EMDB-19256 and PDB-8RK3; (vi) Composite map of bacteriophage JBD30 capsid–neck complex EMDB-19439 and PDB-8RQE; and (vii) Structure of bacteriophage JBD30 baseplate bound to the type IV pilus EMDB-19561. Cryo-EM datasets (Micrographs, Movies, Gain) used for single particle analysis of phage JBD30 were deposited into the Electron Microscopy Public Image Archive (EMPIAR) with the following accession numbers: (i) dataset 1: EMPAIR-12061; (ii) dataset 2: EMPAIR-12062; (iii) dataset 3: EMPIAR-12063; (iv) dataset 4: EMPIAR-12064. Tomograms (raw tilt series and reconstructed tomogram volumes) of JBD30 infected *P. aeruginosa* cells were deposited into EMPIAR with following accession numbers: (i) tomogram of *P. aeruginosa* infected with phage JBD30: EMPIAR-12065 and (ii) tomogram showing the transmembrane channel for JBD30 genome translocation into the *P. aeruginosa* cell: EMPIAR-12066. All custom scripts used during single-particle data reconstruction have been made available at https://github.com/fuzikt/starpy.

The source data of this paper are collected in the following database record: biostudies:S-SCDT-10_1038-S44318-024-00195-1.

## Peer review information

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

## Acknowledgements

We gratefully acknowledge (i) the Cryo-electron Microscopy and Tomography Core Facility and Proteomics Core Facility of the CEITEC MU of CIISB, Instruct-CZ Centre, supported by MEYS CR LM2023042, European Regional Development Fund-Project „UP CIISB" No. CZ.02.1.01/0.0/0.0/18_046/0015974 and e-INFRA CZ (ID:90254) and (ii) the Cellular Imaging Core Facility supported by the Czech-BioImaging large RI project (LM2023050 funded by MEYS CR) for their assistance with obtaining scientific data presented in this paper. Bacteriophage JBD30 was kindly provided by prof. Alan Davidson. Bacterial strain *P. aeruginosa* PAO1 pilA-A86C was provided by prof. Yves V. Brun. We gratefully acknowledge support from the National Institute of Virology and Bacteriology (Program EXCELES, Project ID No. LX22NPO5103) - Funded by the European Union - Next Generation EU. This work received funding from ERC Consolidator Grant No. 101043452 to PP. and from a Brno Ph.D. Talent scholarship funded by Brno city municipality to LV.

## Author contributions

**Lucie Valentová**: Conceptualization; Data curation; Formal analysis; Validation; Investigation; Visualization; Methodology; Writing—original draft; Writing—review and editing. **Tibor Füzik**: Data curation; Formal analysis; Supervision; Validation; Investigation; Visualization; Methodology; Writing—original draft; Writing—review and editing. **Jiří Nováček**: Investigation; Methodology. **Zuzana Hlavenková**: Investigation; Methodology. **Jakub Pospisil**: Formal analysis; Investigation; Methodology. **Pavel Plevka**: Conceptualization; Data curation; Supervision; Funding acquisition; Validation; Writing—original draft; Project administration; Writing—review and editing.

Source data underlying figure panels in this paper may have individual authorship assigned. Where available, figure panel/source data authorship is listed in the following database record: biostudies:S-SCDT-10_1038-S44318-024-00195-1.

## Disclosure and competing interests statement

The authors declare no competing interests.

