## [Peer Review File · The EMBO Journal]

Structure and replication of *Pseudomonas aeruginosa* phage JBD30

Pavel Plevka, Lucie Valentová, Tibor Füzik, Jiri Novacek, Zuzana Hlavenková, and Jakub Pospisil

Corresponding author: Pavel Plevka (pavel.plevka@ceitec.muni.cz)

Review Timeline:

Submission Date:	19th Mar 24
Editorial Decision:	19th Apr 24
Revision Received:	30th May 24
Editorial Decision:	8th Jul 24
Revision Received:	10th Jul 24
Accepted:	24th Jul 24

Editor: Ieva Gailite

Transaction Report:

Dear Dr. Plevka,

Thank you for submitting your manuscript for consideration by the EMBO Journal. We have now received comments from three reviewers, which are included below for your information.

As you can see, all reviewers find the study of interest, while also indicating several aspects in which data presentation, interpretation and description should be improved in the revised manuscript. Based on these generally positive assessments, I would like to invite you to address the comments of all reviewers in a revised version of the manuscript.

I should add that it is The EMBO Journal policy to allow only a single major round of revision and that it is therefore important to resolve the main concerns at this stage. I would also be happy to discuss the revision in more detail via email or phone/videoconferencing.

We generally allow three months as standard revision time, which can be extended if necessary. As a matter of policy, competing manuscripts published during this period will not negatively impact on our assessment of the conceptual advance presented by your study. However, please contact me as soon as possible upon publication of any related work to discuss the appropriate course of action. Should you foresee a problem in meeting this deadline, please let me know in advance to discuss an extension.

When preparing your letter of response to the referees' comments, please bear in mind that this will form part of the Review Process File and will therefore be available online to the community. For more details on our Transparent Editorial Process, please visit our website: <https://www.embopress.org/page/journal/14602075/authorguide#transparentprocess>. Please also see the attached instructions for further guidelines on preparation of the revised manuscript.

Please feel free to contact me if you have any further questions regarding the revision. Thank you for the opportunity to consider your work for publication. I look forward to receiving your revised manuscript.

With best wishes,

Ieva

- a point-by-point response to the referees' comments, with a detailed description of the changes made (as a word file).
- a word file of the manuscript text.
- individual production quality figure files (one file per figure)
- a complete author checklist, which you can download from our author guidelines

(<https://www.embopress.org/page/journal/14602075/authorguide>).

- Expanded View files (replacing Supplementary Information)

We realize that it is difficult to revise to a specific deadline. In the interest of protecting the conceptual advance provided by the work, we recommend a revision within 3 months (18th Jul 2024). Please discuss the revision progress ahead of this time with the editor if you require more time to complete the revisions.

Referee #1:

Review

EMBOJ-2024-117341-T Plevka_P and co-authors

The manuscript "Carpe pili! Hunting strategy, structure, and replica-on of *P. aeruginosa* phage JBD30" by Valentová and co-authors has been submitted to The EMBO Journal. The manuscript reports the structure of siphophage Casadabanvirus JBD30 that uses bacterial pili of host cells at infection. This phage infects the *Pseudomonas aeruginosa* bacteria. Understanding the phage JBD30 organisation and its mechanism of infection through pili opens potential approaches for making medications against *Pseudomonas aeruginosa*. These bacteria are germs that cause infections in the blood, lungs (pneumonia), or other parts of the body after surgery of patients. The authors provided results of structural studies of nearly all phage components at resolutions varying from $\sim 3\text{\AA}$ (for symmetrical elements of the phage) to $8\text{-}10\text{\AA}$ in asymmetrical subcomplexes. The revelation of structural phage elements opens a way for proposing a mechanism of infection. The level of structural analysis is extremely high, and the authors provide significant amount of information on this phage that could be extended and divided nearly into three separate manuscripts (MS). The authors used in their study different approaches of cryo electron microscopy, single particle analysis, helical reconstructions, tomography, and fluorescent microscopy. It would be good to see more details related to the interactions with pili and some biochemical evidence related to the contraction of pili, but apparently it would be a subject for the next study.

In general, the MS is well written, but due to huge amount of information that authors aim to deliver to a reader, and limitation in size imposed on MS at submission some sections of the MS are not easy to follow, and this fact made some impact on the quality of figures. That forced the authors to make figure panels of small sizes. The features discussed in the MS can be seen properly only in electronic presentations (I hope that nobody is using nowadays printed versions of papers since a resolution of figures will be very bad). Possibly, it would be good to make out of this MS two or even three separate papers, where illustrations and descriptions of interactions between proteins will be presented in more details and coherent way. However, it should be done according to the agreement between authors and editors of the journal.

Please find below comments related to this manuscript, that should be addressed. The reviewer hopes that it will help to improve the manuscript and resolve some inaccuracies.

1. Line 378. Possibly it would be more logical to start description of structural analysis of the JBD30 phage starting from procapsids, then full capsids finalising the discussion of results with the empty capsids. Then the authors should consider the neck: the connector complex, followed by tail, baseplate, and fibres. The discussion can be finalised by hypothesis of the process of infectivity based on the found structures. See few other comments related to locations of some paragraphs.
2. It would be useful to have information on the length of homogeneous proteins and where the highest differences between the chains were located. It would be good to know if such differences were linked to the differences in the phage stabilities and interactions between phage components. The authors often mentioned the homology of the proteins but did not discuss the sizes of the chains which were similar.
3. Do the authors have any evidence that tape measure proteins could make a tubular channel within host cellular membrane? Presented tomograms do not indicate that, since they are of a low resolution See other comments in p17.
4. Where are the areas of similarity (groups of aa) of interactions between JBD30 proteins and their corresponding homologs (for example in capsids, portal proteins, or major tail proteins) from other Siphoviridae phages? (see the text)
5. Handedness of the structures. Figure 2. It seems that the panel A is mirrored with respect to the panels B and D. B and D represent classical views of the capsid protein, but in A the structure density has different handedness. In the panel C the tadpole domain is mirrored with respect to one shown in B, that should be clarified. Line 1177. How can it be that the native

- capsid (the empty one) can have mirrored symmetries (with the same I4 sym), the same problem with symmetry C5? The process of DNA packaging into the capsid cannot change its handedness. It looks like two students did different parts of the project and possibly never compared the structures obtained. Both capsids have the same sizes as it shown in figure S2. Question is: which symmetry is correct? How was the structural handedness determined? It seems that should be like in Fig2B and Fig6B that corresponds to the symmetry of the empty capsids shown in the page 37. Moreover, the panels of the procapsids (the overall views) on the page 37 should be mirrored as well. In the figures the surface views of capsids have different handedness and it not always coincide with the handedness of presented atomic ribbon models of the capsid proteins.
6. Line 620. "Particles from each class were manually rotated to orient the
 7. Lines 120-121. It is unclear how "the minor capsid protein may help to increase the capsid size, while it fixes the distances between adjacent capsomers (hexons and pentons)"? What were the differences in the distances between capsid proteins in the phage with and without minor capsid proteins? According to figure 2 the minor capsid proteins work mainly as staplers between capsomers and would not allow any enlargement of the capsid size. Moreover, in figure 2A the capsids have the same size.
 8. Possibly it would be better to put the text in lines 137-147 after line 127.
 9. Lines 154-155. "The continuity of the strands is disrupted below fivefold vertexes the capsid (Fig. S4A)". That is not seen in the figure. The authors have to make that clearer and indicate these disruptions.
 10. Lines 170-172. "A loop from the clip domain intertwines between two neighbouring portal proteins and stabilizes the portal complex (Fig. 3ABC)." This loop is not labelled and not seen in either of these panels of figure 3. Panels C and D are very small. Different conformations of the capsid protein in E are difficult to assess due to similar colours of the overlaid proteins. Figure 3 legend (lines 1070-1071). It was not clear where is N-terminus located since the helices $\alpha 1 - \alpha 3$ are not shown.
 11. Line 181. "Complex the N-terminal whisker (residues 2-14) turns back". This "whisker" is not indicated in the figure. Possibly it would make sense to explain this definition of "whisker" and why this interaction is important.
 12. Section "Interaction of connector complex with DNA" It would be good to see two EM structures of the portal side by side (with and without DNA) to see the difference in locations of the DNA interacting domains of the portal protein and the DNA location withing the portal complex.
 13. Line 256. What are "OB domains"? Please decipher this abbreviation.
 14. It seems that the paragraphs 268-298 will sound better if they will be placed after line 257. Then it will be logical to discuss the fibres of the base plate.
 15. Lines 327. Is it baseplate that binds pili of the bacterium or it should be baseplate fibres? Possibly it would be more accurate if the authors will use the sentence from line 334.
 16. Figure 5F. Did not find any piles in beige, do the authors supposed that it was in light yellow?
 17. Line 392-399. This paragraph should be in the section related to the portal complex since it is related to common features of the portal protein and head completion proteins in phages. The clip domain interacts with the phage capsid in all three forms of the capsid: procapsid, capsids filled with the DNA and in the empty capsids. The clip domain is not shifted out of the capsid, it is located outside in all three forms of the capsids.
 18. Lines 447-449. The paragraph should be extended, since it is unclear what is a "new C-terminus". How many AA were supposed to be cleaved from the TMP? Taking in account that it is highly hydrophobic protein and possibly because of that (I am not sure) it can be attached to membrane, but how the TMP will make a tube? That would require some process that will be energy consuming.
 19. Lines 451-455. Sentences related to "AcrF1 and quorum-sensing antiactivator protein" are very confusing. Possibly it would make sense to describe it in more details. It was not clear what the authors mean that these proteins could prevent "superinfections with other phages"? What the authors mean as "other" phages? The formation of pili was not blocked according to Fig 7 that shows quite a few phages attached to several pili, even several phages could be attached to the pilus fiber.
 20. Figure 7 Insert panel is not visible in the panel C. It would be good to see the 3D structure (from a tomogram) of the area related to the insertion of the phage into the host cell membrane. Panel D -> the procapsid is hardly visible, in the insert; one cannot see any details. Make the figure bigger and indicate the details that the authors would like to show to a reader.
 21. Figure 8 is related to the figure 7. There is confusion with the channels (Fig 7EFG). Can the phage use the channel of the pili to transfer DNA into the host cell? Typically these channels in bacterial cells are rather broad and able to transfer both proteins and nuclear acid in both directions, for example at the conjugation process.
 22. Figure S9. The contrast in all panels is too high, so the fibres are not visible. The arrows point on invisible pili. The authors can improve the contrast by reducing the area corresponding to the cell areas, which are too dense.
 23. Material and methods:
 - a. Did the authors use two different microscopes: one with K2 and another with K3 camera? Or the authors had a rather unique microscope that had two interchangeable cameras under the energy filter?
 - b. What was the point to use energy filters with slits at different voltage?
 - c. How were the microscopes calibrated? What was the difference between them if at smaller magnification 105,000 pixels size on the camera was 0.83 Å (line 563), while at the higher magnification (130,000) the pixel size was 1.06 Å (line 568). At higher magnification the pixel size became smaller and the K2 and K3 cameras have the same size of the sensor.
 - d. What was the point in increasing dose in data set 3 nearly twice compared to data set 1?
 - e. Processing of the capsids. How the authors come to the pixel size 1.08 (line 597) when they processed images the capsids with the decoration protein? Why has it been changed in this type of capsids? Initially the authors have written that data set 1 was collected with the pixel size 1.04 (lines 557-558). The authors have claimed that all magnifications were calibrated. Where does this difference come from?

- f. "Particles from each class were manually rotated to orient the portal " - this sentence is a bit strange, provide more detailed information. It is difficult to believe that images of particles were manually rotated. The portal proteins occupy a unique vertex in phages. They are NOT located at "six distinct vertices " (line 620).
- g. Line 634. Why was the asymmetric reconstruction of the capsid done at the pixel size 0.83?
- h. It would be good if the authors will provide values of B-factors used for sharpening the maps obtained (for all reconstructions), possibly it will make sense to include these values into table S1.
- i. Nearly all sections in the methods have the same phrases related to the software used at the analysis of images, methods of visualisation using Chimera, mask creations, locations of the digital camera in electron microscopes etc. The authors will be able to save some space for the details related to the studies if the repetitions will be moved in one paragraph related to the software and methods used at the analysis of images.

Referee #2:

The MS "Carpe pili! Hunting strategy, structure, and replication of *P. aeruginosa* phage JBD30" by Lucie Valentová et al. describes a high resolution (atomic) structure of the particle of *Pseudomonas aeruginosa* siphophage JBD30, and the imaging of infection of *Pseudomonas* by JBD30 with the help of cryo-electron tomography and fluorescence microscopy.

The MS demonstrates - yet again - that the Plevka group are experts in high resolution cryo-electron microscopy. I can only commend the quality of the high-resolution cryoEM work. I have only a few comments - but they are significant - that concern other parts of the MS.

Contentious point 1. The Abstract.

I believe that apart from the first 2-3 introductory sentences, the Abstract must describe the results of the MS.

17 We show that JBD30 uses its baseplate tail fibres to bind to pili type IV that

18 grow from the poles of *P. aeruginosa* cells.

The statement "we show" appears to be based on the cryoET map shown in Fig. 5F, which does not allow for unambiguous interpretation of the interaction of the pilus with the tail. In addition to the fibers, the pilus appears to interact with other parts of the tail. Hence, "we show" must be accompanied by more definitive experimental data. For example, it could be shown that a fiberless page mutant does not bind to the pilus, or that a recombinantly produced fiber inhibits the interaction between the phage and pilus or the fiber actually binds to the pilus.

18. The pili retraction brings JBD30 to the cell surface.

The MS contains no data that shows this.

18 The

19 structure of the baseplate-pili complex enables the tripod of baseplate receptor binding proteins to
20 attach to the lipopolysaccharides of the outer bacterial membrane.

Again, the attachment of the baseplate-pili complex to the LPS has not been studied in this MS. We see images that show phages bound to the cell surface with their baseplates, but the composition of the attachment points is unknown. Besides LPS, these points can contain outer membrane proteins that are critical for attachments.

20 The tripod and baseplate tip open

21 to release three copies of the tape measure protein, which form a channel through the bacterial cell
22 wall.

The last part of the sentence, that the TMP forms a channel through the bacterial cell wall, has not been shown in this paper.

22 The release of the tail tape measure proteins triggers the DNA ejection.

This has not been shown in this MS.

22 For replication, phage

23 DNA redistributes from the cell poles throughout the cytoplasm.

Even though this is an actual result of this study, it requires clarification (see below).

Contentious point 2. Phage naming nomenclature.

I am very familiar with the *E. coli* phage Mu, which is a contractile tail phage that is famous for its "invertase" enzyme that allows for two different types of tail fibers and tail fiber chaperones be encoded by the same stretch of the genome in opposite directions.

I am extremely confused that the authors call the siphophage studied here a "*Pseudomonas aeruginosa* Mu-like" phage. There is simply no such thing! Digging into the literature reveals that people called some siphophage (or a group of siphophages) Mu-like because they have a similar gene arrangement and a few enzymes. This is beyond bizarre. Many "simple" (like Mu, P2, lambda, etc.) tailed phages have similar gene arrangement and of course (!) they have similar enzymes as those evolve slower

than most structural proteins. Please do not call this phage Mu-like and certainly not *Pseudomonas aeruginosa* Mu-like.

In many places, phage gene products are referred to as "gp35 product". This does not make sense. Gp stands for "gene product". Please, correct throughout.

Contentious point 3, related to L 22-23 of the Abstract "For replication, phage DNA redistributes from the cell poles throughout the cytoplasm".

1. The titer of the phage labeled with DAPI must be compared to that of the unlabeled phage. In some phages DAPI completely blocks the infection. If the DAPI-labeled phage is noninfectious, the results should be interpreted with caution.
2. None of the images clearly demonstrates that the phage DNA actually enters the cell cytoplasm. As far as I can tell, it remains associated with the cell surface and perhaps never enters the cell. Panels in Fig. S10H show puncta that are typical for DAPI-labeled phages. Other figures show huge blobs of DAPI, that may (or may not) correspond to huge phage aggregates, so it is unclear what are we looking at.
3. In Fig. 7F-J and S10 too few images are shown to be convincing to demonstrate that what we are looking at is not a one in a 1000 event.

Referee #3:

This manuscript describes the use of cryoEM and fluorescence microscopy to investigate the structure and infection cycle of phage JBD30. The work is impressive, interesting and timely. The manuscript is also well written and was a pleasure to read. I have no major concerns about the work, but there are some points below that need addressing.

Points to address:

General points:

- "Type IV pili" is more conventional than "pili type IV". I suggest changing this throughout the text and figures.
- Careful with singular and plural e.g. pilus/pili - there are several instances where this should be corrected e.g. on line 19 this should be "baseplate-pilus".
- With regards to the DAPI labelling, could the authors explain how the labelling is maintained during DNA replication? I didn't quite follow how newly produced DNA would be labelled to produce the images seen in Fig. 7IJ & and S10DE.
- The manuscript ends very abruptly with a series of statements. It needs a few concluding sentences to summarise the findings in the wider context.
- There are micrographs shown of full and empty capsids in Fig. 1, but not of the procapsid in Fig. 6, which would be nice to show for completeness. I also couldn't follow how the procapsid sample was obtained - from the data collection parameters in Table S1 it seems to be from a different sample.
- In Methods (centrifugation steps), the "x" is missing from "xg"
- In various cryoEM reconstruction details in Methods, it would be useful to add further explanation about the approach taken to determine the different symmetries that were subsequently applied to various parts of the structure.
- In Table S1, it would be useful to state which dataset number each structure corresponds to, making correlation with the methods section easier.
- I suggest depositing data sets to EMPIAR.

Specific points:

Line 18 - should be "The pilus retraction"

Line 46 - should be "infects the bacterium"

Line 49 - 52 - from these sentences, it is not clear what all of the gene products listed here encode e.g. gp4, gp30, gp35 - is this known? Or is this contained in Table S3, which could be referred to?

Line 103 - should be "of the major"

Line 214 - residues 308-320.. is this the Leu301-Glu325 loop in the legend to Fig. S4? Why the discrepancy?

Line 338 - suggest "Lys68 and Lys11 are predicted to"

Line 441 and Fig. 8 legend title - suggest something more like "working model" rather than "describe the replication cycle"

Line 545 - should be "plunge-frozen"

Line 548 - details of the vitrification for cryoET are missing e.g. freezing parameters, use of fiducials

Line 597 - should be "crYOLO"

Line 1161 - should be "As pili retract"

Line 1164 - should be "form a channel"

Line 1239 - should be "of the T5"

Comments on figures:

In a number of places, the points that are being made in the text are not entirely clear in the figure panels and some different/enlarged views or text indicators would be helpful e.g.

- line 82-83 (symmetry mismatch in Fig. 1)
- line 97 (HK-97 fold in Fig. 2BD)
- line 110 (spine α -helix in Fig. 2B)
- line 111 (hooks interact with each other in Fig. 2B)
- line 181 (N-terminal whisker in Fig. 3E)
- line 232 (decoration domains... protrude tangentially in Fig. 4A)

- line 287-289 - should this point to 5B? It is not apparent in A.
- line 170, 188 - label clip and stem domains in Fig. 3BC
- line 189 - should this point to Fig. 3B?
- line 226 - I take it that the hinge loop in Fig. 4B is the long β -hairpin - suggest using same nomenclature
- Fig. 2 - suggest using a different colour for the cysteines so they stand out
- Fig. 5F - what is the domain in grey? It seems misleading as T4P text label is also shown in grey, but I think this should be beige. It would be helpful to label key domains in colour coded text.
- Fig. 7C - the inset is more of a boxed area - the inset showing a putative channel needs to be enlarged
- Fig. S6 - I couldn't see that BppU was defined, and there are other abbreviations not mentioned in the legend
- Fig. S9 - what does "cell" refer to in the key?

The reviewer's comments are in blue italics, and our responses are in bold black font. Please note that the line numbers in this document refer to the manuscript file and supplementary material file with tracked changes, which were submitted as supplementary files for the revision process.

Referee #1:

Review

EMBOJ-2024-117341-T Plevka_P and co-authors

The manuscript "Carpe pili! Hunting strategy, structure, and replica-on of P. aeruginosa phage JBD30" by Valentová and co-authors has been submitted to The EMBO Journal. The manuscript reports the structure of siphophage Casadabanvirus JBD30 that uses bacterial pili of host cells at infection. This phage infects the Pseudomonas aeruginosa bacteria. Understanding the phage JBD30 organisation and its mechanism of infection through pili opens potential approaches for making medications against Pseudomonas aeruginosa. These bacteria are germs that cause infections in the blood, lungs (pneumonia), or other parts of the body after surgery of patients. The authors provided results of structural studies of nearly all phage components at resolutions varying from $\sim 3\text{\AA}$ (for symmetrical elements of the phage) to $8\text{-}10\text{\AA}$ in asymmetrical subcomplexes. The revelation of structural phage elements opens a way for proposing a mechanism of infection. The level of structural analysis is extremely high, and the authors provide significant amount of information on this phage that could be extended and divided nearly into three separate manuscripts (MS). The authors used in their study different approaches of cryo electron microscopy, single particle analysis, helical reconstructions, tomography, and fluorescent microscopy. It would be good to see more details related to the interactions with pili and some biochemical evidence related to the contraction of pili, but apparently it would be a subject for the next study. In general, the MS is well written, but due to huge amount of information that authors aim to deliver to a reader, and limitation in size imposed on MS at submission some sections of the MS are not easy to follow, and this fact made some impact on the quality of figures. That forced the authors to make figure panels of small sizes. The features discussed in the MS can be seen properly only in electronic presentations (I hope that nobody is using nowadays printed versions of papers since a resolution of figures will be very bad). Possibly, it would be good to make out of this MS two or even three separate papers, where illustrations and descriptions of interactions between proteins will be presented in more details and coherent way. However, it should be done according to the agreement between authors and editors of the journal.

Please find below comments related to this manuscript, that should be addressed. The reviewer hopes that it will help to improve the manuscript and resolve some inaccuracies.

1. Line 378. Possibly it would be more logical to start description of structural analysis of the JBD30 phage starting from procapsids, then full capsids finalising the discussion of results with the empty capsids. Then the authors should consider the neck: the connector complex, followed by tail, baseplate, and fibres. The discussion can be finalised by hypothesis of the process of infectivity based on the found structures. See few other comments related to locations of some paragraphs.

A: We appreciate the logic of the manuscript reorganization suggested by reviewer #1. However, we prefer to retain the current manuscript flow that follows the phage infection cycle: It starts with describing the structure of JBD30 virion, followed by phage-pilus interaction, genome delivery, and concludes with assembly of progeny procapsids.

2. It would be useful to have information on the length of homogeneous proteins and where the highest differences between the chains were located. It would be good to know if such differences were linked to the differences in the phage stabilities and interactions between phage components. The authors often mentioned the homology of the proteins but did not discuss the sizes of the chains which were similar.

A: Sizes of the homologous proteins and sequence identities compared to those of JBD30 are listed in Table S3. There is not enough information about the compared proteins to be able to relate the differences in sequences to the stability of phage particles. We have now included additional columns in the table showing the number of residues in each of the compared proteins.

3. Do the authors have any evidence that tape measure proteins could make a tubular channel within host cellular membrane? Presented tomograms do not indicate that, since they are of a low resolution See other comments in p17.

A: Section from a tomogram shown in Fig. 7 depicts a putative channel through the *P. aeruginosa* cell wall. However, we agree that we do not provide experimental evidence that the tape measure protein forms a channel through the host membrane. We have now removed this speculation from the abstract, and in the results and discussion section, we state that we make this speculation based on previously published results (lines 418-419): "The rest of the tape measure protein (residues 57–929) was predicted to form a transmembrane channel composed of α -helical segments (Fig. S7C) (Mistry et al., 2021)."

4. Where are the areas of similarity (groups of aa) of interactions between JBD30 proteins and their corresponding homologs (for example in capsids, portal proteins, or major tail proteins) from other Siphoviridae phages? (see the text)

A: We are sorry; we do not understand this question/comment.

5. Handedness of the structures. Figure 2. It seems that the panel A is mirrored with respect to the panels B and D. B and D represent classical views of the capsid protein, but in A the structure density has different handedness. In the panel C the tadpole domain is mirrored with respect to one shown in B, that should be clarified.

A: The panels of Fig. 2 are not mirrored. We have now extended the figure legend to explain the orientation of the structures in detail (lines 1,218-1,219):

"(D) Cartoon representation of JBD30 minor capsid protein coloured according to domain composition and viewed from the capsid. "

Line 1177. How can it be that the native capsid (the empty one) can have mirrored symmetries (with the same I4 sym), the same problem with symmetry C5? The process of DNA packaging into the capsid cannot change its handedness. It looks like two students did different parts of the project and possibly never compared the structures obtained. Both capsids have the same sizes as it shown in figure S2. Question is: which symmetry is correct? How was the structural handedness determined? It seems that should be like in Fig2B and

Fig. 6B that corresponds to the symmetry of the empty capsids shown in the page 37. Moreover, the panels of the procapsids (the overall views) on the page 37 should be mirrored as well. In the figures the surface views of capsids have different handedness and it not always coincide with the handedness of presented atomic ribbon models of the capsid proteins.

A: Thank you; indeed the handedness of the structures was mixed and wrong. We have now corrected the handedness of the empty capsid reconstructed with imposed icosahedral symmetry and the empty particles reconstructed with imposed C5 symmetry in Fig. S2. We overlooked the handedness because these maps were not used for PDB model building. Because of the high resolution of the reconstructions, we can determine their handedness by fitting the structure of major capsid protein into them.

6. Line 620. "Particles from each class were manually rotated to orient the

A: We have now modified the sentence (lines 726-727): "Particles from each class were rotated to orient the portal complexes along the Z-axis using RELION 3.1 (Zivanov et al., 2018) and custom-written script `math_star.py` (GitHub/fuzikt)."

7. Lines 120-121. It is unclear how "the minor capsid protein may help to increase the capsid size, while it fixes the distances between adjacent capsomers (hexons and pentons)"? What were the differences in the distances between capsid proteins in the phage with and without minor capsid proteins? According to figure 2 the minor capsid proteins work mainly as staplers between capsomers and would not allow any enlargement of the capsid size. Moreover, in figure 2A the capsids have the same size.

A: We were referring to the putative function of minor capsid proteins in increasing the size of phage heads of other bacteriophages. We have now re-written the sentence to avoid the confusion (lines 130-134):

"Minor capsid proteins of other phages have been shown to: stabilize the capsid architecture against internal genome pressure (T4 Soc, YSD1 gp16) (Qin et al., 2010; Hardy et al., 2020), help to increase the capsid size and volume (P23-45 gp88) (Bayfield et al., 2019), play a role in target cell recognition (T5 pb10, T4 Hoc) (Vernhes et al., 2017; Sathaliyawala et al., 2010), and participate in virion assembly (λ gpD) (Lander et al., 2008)."

8. Possibly it would be better to put the text in lines 137-147 after line 127.

A: Text in lines 128-136 of the first version of the manuscript describes the structure and properties of the minor capsid protein, whereas the text in lines 137-147 compares its properties to those of other bacteriophages. We prefer not to change the order of the text.

9. Lines 154-155. "The continuity of the strands is disrupted below fivefold vertexes of the capsid (Fig. S4A)". That is not seen in the figure. The authors have to make that clearer and indicate these disruptions.

A: We have now included pentagons indicating positions of fivefold vertices in Fig. S4A to show that the density is not structured into strands in their vicinity (Supplementary data line 59):

"Positions of selected fivefold vertices are indicated by purple pentagons."

10. Lines 170-172. "A loop from the clip domain intertwines between two neighbouring portal proteins and stabilizes the portal complex (Fig. 3ABC)." This loop is not labelled and not seen in either of these panels of figure 3. Panels C and D are very small.

A: We have now colored the loops in green and indicated their positions by labels in Fig. 3C.

Different conformations of the capsid protein in E are difficult to assess due to similar colours of the overlaid proteins.

A: We have now changed the color of the capsid proteins to make them easily distinguishable (lines 1,251-1,255):

"(E) Superimposition of cartoon representations of major capsid protein interacting with another capsomer (beige and N-terminus in dark blue) and capsid protein adjacent to portal dodecamer (purple and N-terminus in light blue) and one showing the bending of the N-terminal arm which enables incorporation of the portal complex into the capsid."

Figure 3 legend (lines 1070-1071). It was not clear where is N-terminus located since the helices $\alpha 1 - \alpha 3$ are not shown.

A: Thank you. We have now included labels for helices $\alpha 1 - \alpha 3$.

11. Line 181. "Complex the N-terminal whisker (residues 2-14) turns back". This "whisker" is not indicated in the figure. Possibly it would make sense to explain this definition of "whisker" and why this interaction is important.

A: We have now highlighted the whisker in Fig. 3E, and we have removed the term "whisker" from the text (lines 196-200):

"In contrast, at the interface with the portal complex, the N-terminus of the major capsid protein (residues 2–14) turns back and forms a short α -helix positioned parallel to the extended loop of the neighboring major capsid protein, thus leaving more space around the fivefold axis for the portal complex (Fig. 3E).

12. Section "Interaction of connector complex with DNA" It would be good to see two EM structures of the portal side by side (with and without DNA) to see the difference in locations of the DNA interacting domains of the portal protein and the DNA location within the portal complex.

A: We have now modified Fig. 3B to show the details requested (lines 1,237-1,241):

"(B) Portals of virion and empty particle differ in the structure of wing domain helix $\alpha 12$. Cartoon representation of two opposite portal proteins from a virion coloured according to domains (as in panel (A)). Portal proteins from an empty particle are depicted in grey. Helix $\alpha 12$ is fully stretched in the empty particle and the tunnel loop narrows the portal channel to 21 Å."

13. Line 256. What are "OB domains"? Please decipher this abbreviation.

A: OB stands for oligosaccharide binding. We have now included an explanation of the abbreviation (lines 1,273):

"OB domain stands for oligosaccharide binding domain."

14. It seems that the paragraphs 268-298 will sound better if they will be placed after line 257. Then it will be logical to discuss the fibres of the base plate.

A: We have re-arranged the text as requested by the reviewer.

15. Lines 327. Is it baseplate that binds pili of the bacterium or it should be baseplate fibres? Possibly it would be more accurate if the authors will use the sentence from line 334.

A: We have modified the title to (lines 351): Mechanism of JBD30 binding to type IV pilus

16. Figure 5F. Did not find any piles in beige, do the authors supposed that it was in light yellow?

A: To avoid confusion in color naming, we have now included the label of the pilus in Fig. 5F.

17. Line 392-399. This paragraph should be in the section related to the portal complex since it is related to common features of the portal protein and head completion proteins in phages.

A: We prefer not to re-organize the text since the paragraph compares the structures of the portal in procapsid and capsid.

The clip domain interacts with the phage capsid in all three forms of the capsid: procapsid, capsids filled with the DNA and in the empty capsids. The clip domain is not shifted out of the capsid, it is located outside in all three forms of the capsids.

A: The clip domain of JBD30 does not interact with the capsid of the virion. The closest distance between any atoms of capsid protein and the clip domain is 18 Å (Please see Fig. 6E). We have now modified the description to avoid the misleading statement about the clip domain shifting out of the capsid (lines 446-448) :

"Unlike in the procapsid, in the JBD30 virion, the stem domain helices $\alpha 9$ and $\alpha 11$ cross the capsid shell (Fig. 6E). The clip domain reaches further out of the capsid to enable the attachment of the adaptor complex (Fig. 3A, 6E)."

18. Lines 447-449. The paragraph should be extended, since it is unclear what is a "new C-terminus". How many AA were supposed to be cleaved from the TMP? Taking in account that it is highly hydrophobic protein and possibly because of that (I am not sure) it can be attached to membrane, but how the TMP will make a tube? That would require some process that will be energy consuming.

A: We have now included references in the indicated sentences and expanded the explanation for our speculation of the channel formation (lines 498-505):

"The baseplate tip opens, the C-terminal domain of tape measure protein (residues 1,067 – 1,158) is extruded from the baseplate, and the protein is cleaved at residue 1,066 (Linares et al., 2022). We hypothesize that the new C-terminus of the tape measure protein binds to the baseplate core (Fig. S7). The rest of the tape measure protein, containing 78% amino acids with hydrophobic or neutral sidechains, is expelled from the tail tube, inserts into the membrane, and re-folds into a new conformation to form a channel for genome delivery into the host cytoplasm (Fig. 7C, Fig. S7) (Boulanger et al., 2008; Mahony et al., 2016)."

19. Lines 451-455. Sentences related to "AcrF1 and quorum-sensing antiactivator protein" are very confusing. Possibly it would make sense to describe it in more details. It was not clear what the authors mean that these proteins could prevent "superinfections with other

phages "? What the authors mean as "other" phages? The formation of pili was not blocked according to Fig 7 that shows quite a few phages attached to several pili, even several phages could be attached to the pilus fiber.

A: We have now re-written the sentences (lines 509-516):

"JBD30 then employs anti-CRISPR protein AcrF1 to evade the host immune system (Bondy-Denomy et al., 2013) and quorum-sensing anti-activator protein Aqs1 to attenuate cell-to-cell communication, which would result in reduced pili production (Table S1) (Shah et al., 2021). Based on the lysis-lysogeny decision controlled by the gp1, λ -like repressor cl, JBD30 DNA either integrates into the host chromosome (using gp6 encoded transposase) or enters the lytic cycle to produce new virions (Table S1)."

Fig. 7A and Fig. S9 show *P. aeruginosa* cells with pili and bound phages; in Fig. 7A there are indeed several phages attached to the same pilus. However, these samples were plunge-frozen 2-10 minutes post-infection. The infecting phages have not had enough time to initiate expression of Aqs1 since it has been shown by Shah et al. that the transcription of Aqs1 starts 10 minutes post-infection.

Shah M, Taylor VL, Bona D, Tsao Y, Stanley SY, Pimentel-Elardo SM, McCallum M, Bondy-Denomy J, Howell PL, Nodwell JR, Davidson AR, Moraes TF, Maxwell KL. A phage-encoded anti-activator inhibits quorum sensing in *Pseudomonas aeruginosa*. *Mol Cell*. 2021 Feb 4;81(3):571-583.e6. doi: 10.1016/j.molcel.2020.12.011. Epub 2021 Jan 6. PMID: 33412111.

20. Figure 7 Inset panel is not visible in the panel C. It would be good to see the 3D structure (from a tomogram) of the area related to the insertion of the phage into the host cell membrane.

A: We have now included 2D overlay segmentation in panel C and an enlarged inset showing the putative transmembrane channel spanning the *P. aeruginosa* cell wall. We prefer not to show the cryo-EM density of the tomogram since it is noisy and suffers from a missing wedge artifact. We deposited the tomogram into EMPIAR as entry #12066, and it is now available for inspection.

Panel D -> the procapsid is hardly visible, in the inset; one cannot see any details. Make the figure bigger and indicate the details that the authors would like to show to a reader.

A: We have now included boxes highlighting the positions of all proheads in panel D and an additional inset in which the procapsid is magnified. Our objective is not to show details of the procapsids, which are well resolved in the single-particle reconstruction.

21. Figure 8 is related to the figure 7. There is confusion with the channels (Fig 8EFG). Can the phage use the channel of the pili to transfer DNA into the host cell? Typically these channels in bacterial cells are rather broad and able to transfer both proteins and nuclear acid in both directions, for example at the conjugation process.

A: Type IV pilus is a thin fiber with an outer diameter of 5 nm and no inner channel into which the phage could eject its DNA. Type IV pili do not serve in the conjugation process. We have now updated the schematic illustration of type IV pilus in the scheme in Fig. 7 so that it does not misleadingly indicate that type IV pilus assembly machine forms a pore through the inner bacterial membrane.

Wang, F. et al. Cryoelectron microscopy reconstructions of the *Pseudomonas aeruginosa* and *Neisseria gonorrhoeae* type IV pili at sub-nanometer resolution. *Structure* 25, 1423–1435 (2017).

Craig, L., Forest, K.T. & Maier, B. Type IV pili: dynamics, biophysics and functional consequences. *Nat Rev Microbiol* 17, 429–440 (2019). <https://doi.org/10.1038/s41579-019-0195-4>

22. Figure S9. The contrast in all panels is too high, so the fibres are not visible. The arrows point on invisible pili. The authors can improve the contrast by reducing the area corresponding to the cell areas, which are too dense.

A: Thank you. As suggested, we have now cropped the images, increased contrast, repositioned the arrows to point directly to pili, and exported the images with higher resolution.

23. Material and methods:

a. Did the authors use two different microscopes: one with K2 and another with K3 camera? Or the authors had a rather unique microscope that had two interchangeable cameras under the energy filter?

A: The images were collected over a period of time during which the microscope was upgraded to a K3 camera.

b. What was the point to use energy filters with slits at different voltage?

A: We used 20eV slit width with a K2 camera due to the lower stability of the Quantum K2 imaging system. The K3 system has higher stability, and, therefore, we used a 10eV slit for imaging. We used 50eV slit width to collect low-magnification polygon montages of large grid areas.

c. How were the microscopes calibrated? What was the difference between them if at smaller magnification 105,000 pixels size on the camera was 0.83 Å (line 563), while at the higher magnification (130,000) the pixel size was 1.06 Å (line 568). At higher magnification the pixel size became smaller and the K2 and K3 cameras have the same size of the sensor.

A: The discrepancy between magnifications and pixel sizes, despite the same sizes of detector sensors of K2 and K3, is caused by differences in the imaging filter optics. The pixel sizes were calibrated using Apoferritin samples.

d. What was the point in increasing dose in data set 3 nearly twice compared to data set 1?

A: Both the datasets were recorded using K2, but the method of saving images was upgraded between the data collections. When collecting the first dataset, we could save only 25 frame-movies, while for the third dataset, the K2 was "hacked," and we could save 40 frame-movies and thus could use a higher dose. We used MotionCor2, which enabled us to omit high-resolution information from frames collected later during the exposition (dose-weighted low-pass filtering).

e. Processing of the capsids. How the authors come to the pixel size 1.08 (line 597) when they processed images the capsids with the decoration protein? Why has it been changed in this type of capsids? Initially the authors have written that data set 1 was collected with the pixel size 1.04 (lines 557-558). The authors have claimed that all magnifications were calibrated. Where does this difference come from?

A: We apologize; the 1.08 value was a typo. This has now been corrected to 1.04.

f. "Particles from each class were manually rotated to orient the portal" - this sentence is a bit strange, provide more detailed information. It is difficult to believe that images of particles were manually rotated.

A: We have now expanded the description of the process (lines 717-727):

"Images of particles with orientations assigned from the final icosahedral 3D refinement were downsampled and re-extracted (256 × 256 px, 2.73525 Å/px) and aligned to have the connectors oriented along the z-axis using several rounds of 2D classification with masks (created in FIJI (Schindelin et al, 2012)) that only included the space for the connector. The orientation search was omitted during the classifications, and particle orientations and shifts were taken from the icosahedral 3D refinement. Because of the embedding in a thin layer of vitreous ice, the JBD30 particles were oriented with their tails pointing along the projection plane. Therefore, the classification resulted in classes of particles with the portal complexes oriented along one of the six directions of fivefold vertices from the icosahedral alignment, deviating the least from the projection plane. Particles from each class were rotated to orient the portal complexes along the Z-axis using RELION 3.1 (Zivanov et al., 2018) and custom-written script math_star.py (GitHub/fuzikt)."

The portal proteins occupy a unique vertex in phages. They are NOT located at "six distinct vertices" (line 620).

A: Thank you. We have now re-written the explanation (lines 717-734):

Images of particles with orientations assigned from the final icosahedral 3D refinement were downsampled and re-extracted (256 × 256 px, 2.73525 Å/px) and aligned to have the connectors oriented along the z-axis using several rounds of 2D classification with masks (created in FIJI (Schindelin et al, 2012)) that only included the space for the connector. The orientation search was omitted during the classifications, and particle orientations and shifts were taken from the icosahedral 3D refinement. Because of the embedding in a thin layer of vitreous ice, the JBD30 particles were oriented with their tails pointing along the projection plane. Therefore, the classification resulted in classes of particles with the portal complexes oriented along one of the six directions of fivefold vertices from the icosahedral alignment deviating the least from the projection plane. Particles from each class were rotated to orient the portal complexes along the Z-axis using RELION 3.1 (Zivanov et al., 2018) and custom-written script math_star.py (GitHub/fuzikt). The map from the icosahedral 3D refinement was rescaled (relion_image_handler) and used as an initial model for the first 3D refinement of the capsid in C5 symmetry, where only local searches of the Euler rot angle were allowed. A mask that included the capsid and portal was created using the programs UCSF Chimera (Pettersen et al., 2004) and relion_mask_create (Scheres, 2012). Subsequent steps of 3D refinement, CTF refinement, and Ewald sphere correction (Zivanov et al., 2018) were done using unbinned data (840 × 840 px, 0.8336 Å/px). The final map was threshold-masked, divided by the modulation transfer function, and B-factor sharpened during post-processing.

g. Line 634. Why was the asymmetric reconstruction of the capsid done at the pixel size 0.83?

A: We used data collected using a K3 detector for this reconstruction.

h. It would be good if the authors will provide values of B-factors used for sharpening the maps obtained (for all reconstructions), possibly it will make sense to include these values into table S1.

A: Thank you, we have now included the B-factor values (Table S2).

i. Nearly all sections in the methods have the same phrases related to the software used at the analysis of images, methods of visualisation using Chimera, mask creations, locations of the digital camera in electron microscopes etc. The authors will be able to save some space for the details related to the studies if the repetitions will be moved in one paragraph related to the software and methods used at the analysis of images.

A: We apologize for the repetitive texts, but they cannot be easily combined because there are differences, and describing the reconstruction procedures in a combined way would make the text incomprehensible.

Referee #2:

The MS "Carpe pili! Hunting strategy, structure, and replication of P. aeruginosa phage JBD30" by Lucie Valentová et al. describes a high resolution (atomic) structure of the particle of Pseudomonas aeruginosa siphophage JBD30, and the imaging of infection of Pseudomonas by JBD30 with the help of cryo-electron tomography and fluorescence microscopy.

The MS demonstrates - yet again - that the Plevka group are experts in high resolution cryo-electron microscopy. I can only commend the quality of the high-resolution cryoEM work. I have only a few comments - but they are significant - that concern other parts of the MS.

Contentious point 1. The Abstract.

I believe that apart from the first 2-3 introductory sentences, the Abstract must describe the results of the MS.

We show that JBD30 uses its baseplate tail fibres to bind to pili type IV that grow from the poles of P. aeruginosa cells.

The statement "we show" appears to be based on the cryoET map shown in Fig. 5F, which does not allow for unambiguous interpretation of the interaction of the pilus with the tail. In addition to the fibers, the pilus appears to interact with other parts of the tail. Hence, "we show" must be accompanied by more definitive experimental data. For example, it could be shown that a fiberless page mutant does not bind to the pilus, or that a recombinantly produced fiber inhibits the interaction between the phage and pilus or the fiber actually binds to the pilus.

A: We have now modified the abstract and manuscript text to avoid the statement that JBD30 binding to pili is exclusively mediated by tail fibers (lines 15-17):

"Here we present the infection cycle of siphophage Casadabanvirus JBD30, which uses its baseplate to bind to the type IV pilus of *Pseudomonas aeruginosa*."

And lines 367-371:

"The receptor binding protein of JBD30 forms a putative contact with the type IV pilus (Fig. 5F). This interaction might help to stabilize the baseplate in an orientation along the pilus towards the bacterial cell surface. However, no interaction between the receptor binding protein and major pilin protein pilA was predicted using Alphafold2 multimer (Evans et al., 2021)."

We appreciate that reviewer #2 suggested additional experiments to determine the roles of JBD30 tail proteins in pilus attachment; however, the experiments may be impossible to perform or challenging to interpret. It has been shown by Harvey et al. 2018 that JBD30 requires type IV pili for infection. Therefore, engineered phages without fibers may not be viable. Furthermore, gp47 and gp48 form a three-protomer long segment of a helical structure in the JBD30 baseplate. Thus, recombinant expression of gp47 and gp48 is likely to result in the production of fibers, which would be difficult to use to characterize their binding to pili. Considering these challenges, we would like to postpone the proposed experiments for a follow-up manuscript.

Harvey H, Bondy-Denomy J, Marquis H, Sztanko KM, Davidson AR, Burrows LL.

***Pseudomonas aeruginosa* defends against phages through type IV pilus glycosylation. Nat**

Microbiol. 2018 Jan;3(1):47-52. doi: 10.1038/s41564-017-0061-y. Epub 2017 Nov 13. PMID: 29133883.

18. The pili retraction brings JBD30 to the cell surface.

The MS contains no data that shows this.

We have now included additional experiments to provide evidence that JBD30 is brought to a cell surface by pilus retraction (Fig. S13 and lines 376-387):

"JBD30 attachment to the type IV pilus enables two possibilities for how the phage can reach the cell surface. One-dimensional diffusion of a phage particle along the pilus may bring it to the cell surface. Alternatively, the phage may bind to one segment of the pilus and be carried to the cell surface by pilus retraction. To differentiate between the two alternatives, we added JBD30 to bacterial cells incubated at 4°C. At this low temperature phage particles attached to pili, but most of them did not reach the cell surface (Fig. S13). After heating the cells to 37°C, phage particles were brought to the cell surface. Lowering the temperature from 37°C to 4°C only causes a 10% reduction in the diffusion rate (Berg, 1993). However, cooling to 4°C reduces *P. aeruginosa* metabolic activity and the pili polymerization and retraction dynamics (Tsuji et al., 1982; van der Wielen et al., 2023; Schneider & Doetsch, 1977). Since the low temperature prevented the movement of phages along pili, we propose that JBD30 particles are brought to the cell surface by pili retraction."

The structure of the baseplate-pili complex enables the tripod of baseplate receptor binding proteins to attach to the lipopolysaccharides of the outer bacterial membrane.

Again, the attachment of the baseplate-pili complex to the LPS has not been studied in this MS. We see images that show phages bound to the cell surface with their baseplates, but the composition of the attachment points is unknown. Besides LPS, these points can contain outer membrane proteins that are critical for attachments.

A: We agree; we have now modified the abstract and manuscript text to remove this claim (lines 18-19):

"The structure of the baseplate-pilus complex enables the tripod of baseplate receptor binding proteins to attach to the outer bacterial membrane."

Lines 1,330-1,331:

"As pilus retracts, phage particle is brought to cell surface and binds it using receptor binding proteins."

The tripod and baseplate tip open to release three copies of the tape measure protein, which form a channel through the bacterial cell wall.

The last part of the sentence, that the TMP forms a channel through the bacterial cell wall, has not been shown in this paper.

A: We agree; we have now modified the abstract to remove this claim (lines 20-21):

"The tripod and baseplate tip open to release three copies of the tape measure protein, which is followed by the DNA ejection."

The release of the tail tape measure proteins triggers the DNA ejection.

This has not been shown in this MS.

A: We agree; we have now modified the abstract to remove this claim (lines 20-21):

"The tripod and baseplate tip open to release three copies of the tape measure protein, which is followed by the DNA ejection."

For replication, phage DNA redistributes from the cell poles throughout the cytoplasm. Even though this is an actual result of this study, it requires clarification (see below).

A: As part of the shortening of the abstract to 175 words, this sentence has been removed.

Contentious point 2. Phage naming nomenclature.

I am very familiar with the E. coli phage Mu, which is a contractile tail phage that is famous for its "invertase" enzyme that allows for two different types of tail fibers and tail fiber chaperones be encoded by the same stretch of the genome in opposite directions.

I am extremely confused that the authors call the siphophage studied here a "Pseudomonas aeruginosa Mu-like" phage. There is simply no such thing! Digging into the literature reveals that people called some siphophage (or a group of siphophages) Mu-like because they have a similar gene arrangement and a few enzymes. This is beyond bizarre. Many "simple" (like Mu, P2, lambda, etc.) tailed phages have similar gene arrangement and of course (!) they have similar enzymes as those evolve slower than most structural proteins. Please do not call this phage Mu-like and certainly not Pseudomonas aeruginosa Mu-like.

A: Thank you. We have now removed all instances of mentioning phage Mu.

In many places, phage gene products are referred to as "gp35 product". This does not make sense. Gp stands for "gene product". Please, correct throughout.

A: Thank you. We have now corrected the manuscript as requested.

Contentious point 3, related to L 22-23 of the Abstract "For replication, phage DNA redistributes from the cell poles throughout the cytoplasm".

1. The titer of the phage labeled with DAPI must be compared to that of the unlabeled phage. In some phages DAPI completely blocks the infection. If the DAPI-labeled phage is noninfectious, the results should be interpreted with caution.

A: We have now included additional results in Fig. S11 showing that DAPI has no or only a limited effect on JBD30.

2. None of the images clearly demonstrates that the phage DNA actually enters the cell cytoplasm. As far as I can tell, it remains associated with the cell surface and perhaps never enters the cell. Panels in Fig. S10H show puncta that are typical for DAPI-labeled phages. Other figures show huge blobs of DAPI, that may (or may not) correspond to huge phage aggregates, so it is unclear what are we looking at.

A: We have now included xz and yz slices through the 3D reconstructions of z-stacks of fluorescent images of P. aeruginosa cells infected by DAPI-labelled JBD30 in Fig. S12. The images show the labeled DNA inside the cells.

3. In Fig. 7F-J and S10 too few images are shown to be convincing to demonstrate that what we are looking at is not a one in a 1000 event.

A: We have now included five additional examples of infected cells in Fig. S12. Upon request, we will happily provide a complete dataset for inspection.

Referee #3:

This manuscript describes the use of cryoEM and fluorescence microscopy to investigate the structure and infection cycle of phage JBD30. The work is impressive, interesting and timely. The manuscript is also well written and was a pleasure to read. I have no major concerns about the work, but there are some points below that need addressing.

Points to address:

General points:

- "Type IV pili" is more conventional than "pili type IV". I suggest changing this throughout the text and figures.

A: Thank you. We have now corrected all instances to the more common word order.

- Careful with singular and plural e.g. pilus/pili - there are several instances where this should be corrected e.g. on line 19 this should be "baseplate-pilus".

A: Thank you. We have now checked and corrected the manuscript.

- With regards to the DAPI labelling, could the authors explain how the labelling is maintained during DNA replication? I didn't quite follow how newly produced DNA would be labelled to produce the images seen in Fig. 7IJ & and S10DE.

A: The DAPI from labeled phages remained in the infected cells and, therefore, became incorporated into progeny virions. We have now included this explanation in the manuscript (lines 1,324-1,325):

"The DAPI stain from the infecting phage particles remained in the cell and in the late stages of infection re-distributed to the newly synthesized DNA."

- The manuscript ends very abruptly with a series of statements. It needs a few concluding sentences to summarise the findings in the wider context.

A: We have now included concluding sentences (lines 537-540):

"The combination of cryo-electron tomography and fluorescent microscopy has enabled characterization of the replication cycle of phage JBD30, shedding light on its intricate mechanisms of host cell recognition, genome delivery, and progeny particle formation.

- There are micrographs shown of full and empty capsids in Fig. 1, but not of the procapsid in Fig. 6, which would be nice to show for completeness. I also couldn't follow how the procapsid sample was obtained - from the data collection parameters in Table S1 it seems to be from a different sample.

A: We have now included an example micrograph of a procapsid in Fig. 6A as an inset. As for the sample preparation – we were lucky, and the purified phage sample also contained procapsids. We have now included this information in the manuscript (lines 425-427):

"The sample of JBD30 used for the collection of electron micrographs contained a fraction of procapsids, which enabled their structure determination (Fig. S1, Table S2).

- In Methods (centrifugation steps), the "x" is missing from "xg"

A: Thank you, this has now been corrected:

Line 548: "The cell debris were removed by centrifugation at 5,000 × g at 4 °C for 20 min"

Lines 623-624: "Two minutes post infection, the cells were pelleted (1 min, 5,000 × g) to increase the cell density and remove the non-adsorbed phages."

Lines 552-553: "The phage lysate (500 ml) was centrifuged in a 50.2 Ti rotor (Beckman Coulter) at 54,000 × g and 10°C for 2.5 h (Optima XPN-80 Ultracentrifuge, Beckman Coulter)."

Lines 588-590: "Labelled phage particles were pelleted by ultracentrifugation (54,000 × g, 2.5 h), resuspended in 2 ml of phage buffer and pelleted again."

Lines 564-566: "After 10 min of incubation at 37 °C and 100 RPM to allow the adsorption of phages to bacterial cells, the sample was centrifuged (14,000 × g, 90 s, RT) and resuspended in 1 ml of LB medium, then centrifuged again and resuspended in 1 ml of LB medium. "

Lines 593-594: "Cells from 1 ml of the culture were harvested by centrifugation (1 min, 8,000 × g) and resuspended in 500 µl of PBS buffer."

Lines 596-597: "To remove the unbound fluorescent dye, the labelled cells were centrifuged (1 min, 8,000 × g) and resuspended in fresh 500 µl of PBS."

- In various cryoEM reconstruction details in Methods, it would be useful to add further explanation about the approach taken to determine the different symmetries that were subsequently applied to various parts of the structure.

A: We have based our decision on symmetries of homologous complexes from other phages. The threefold symmetry of the baseplate was apparent from reference-free 2D class averages. We have now included this information in the manuscript (lines 822-823): "Some of the 2D class averages of the baseplates exhibited threefold symmetry, which was subsequently imposed during the 3D reconstruction process."

- In Table S1, it would be useful to state which dataset number each structure corresponds to, making correlation with the methods section easier.

A: Thank you, we have now included this information in Table S2 (new table order).

- I suggest depositing data sets to EMPIAR.

A: We have now deposited data to EMPAIR and included deposition numbers in Table S2 (new table order).

Specific points:

Line 18 - should be "The pilus retraction"

A: As part of the shortening of the abstract to 175 words, this sentence has been removed.

Line 46 - should be "infects the bacterium"

A: Thank you, this has now been corrected (lines 46-47):

"Bacteriophage Casadabanvirus JBD30, from the order Caudoviricetes, is a temperate phage that infects the bacterium *Pseudomonas aeruginosa*. (Bondy-Denomy et al., 2016)."

Line 49 - 52 - from these sentences, it is not clear what all of the gene products listed here encode e.g. gp4, gp30, gp35 - is this known? Or is this contained in Table S3, which could be referred to?

A: Thank you, we have now included a reference to the table (now Table S1).

Line 103 - should be "of the major"

A: Thank you, this has now been corrected (lines 113-114).

"The extended loop of the major capsid protein stretches over the peripheral domain of the neighbouring subunit positioned clockwise in the same hexamer or pentamer (Fig. 2B).

Line 214 - residues 308-320.. is this the Leu301-Glu325 loop in the legend to Fig. S4? Why the discrepancy?

A: Thank you, we have now removed the discrepancy (lines 234-236):

"Whereas most of the portal protein structure exhibits twelfold symmetry, the structures of tunnel loops (residues 301–LGGTLTSTTSQSGGGAFALGQVHNE–325) differ between the individual subunits of the complex (Fig. S4F)."

Line 338 - suggest "Lys68 and Lys11 are predicted to"

A: Thank you, this has now been corrected (lines 364-366):

"Side chains of pilA residues Lys68 and Lys112 are predicted to interact with Asp185 of gp47 and Tyr169 of gp48, respectively (Fig. S6K)."

Line 441 and Fig. 8 legend title - suggest something more like "working model" rather than "describe the replication cycle"

A: We have now re-written the text (lines 493-494):

"The combination of data from cryo-electron tomography and fluorescent microscopy enabled us to propose a working model of the replication cycle of JBD30 (Fig. 8)."

Line 1336:

"Fig. 8. Structure and proposed model of replication cycle of JBD30."

Line 545 - should be "plunge-frozen"

A: Thank you, this has now been corrected (lines 618-620):

"Grids were blotted (blotting force 0, blotting time 2 s, 100% humidity, waiting time 15 s), plunge-frozen in liquid ethane using a Vitrobot Mark IV, and stored in liquid nitrogen."

Line 548 - details of the vitrification for cryoET are missing e.g. freezing parameters, use of fiducials

A: Thank you, this has now been completed (lines 616-630):

"A sample with purified JBD30 (4 µl of 1011 PFU/ml) was applied onto a Quantifoil™ grid (2/1, Cu, mesh 300) glow-discharged in H₂O plasma using a Gatan Solarus II. Grids were blotted (blotting force 0, blotting time 2 s, 100% humidity, waiting time 15 s), plunge-frozen in liquid ethane using a Vitrobot Mark IV, and stored in liquid nitrogen. *P. aeruginosa* cells for cryo-electron tomography sample preparation were grown in LB

medium at 37 °C, 250 RPM to OD600 ≈ 0.3 (108CFU/ml). At time 0, the phage lysate was added to the culture at a multiplicity of infection of 500. The infected culture was incubated at 30 °C, 60 RPM. Two minutes post infection, the cells were pelleted (1 min, 5,000 × g) to increase the cell density and remove the non-adsorbed phages. The pellet was resuspended in fresh LB medium to reach OD600 ≈ 9. Samples for vitrification were taken at distinct time points over the course of infection. A volume of 4 μl of the infected cells was applied onto a Quantifoil™ grid (2/1, Cu, mesh 300) glow-discharged in H₂O plasma using a Gatan Solarus II. Before sample application, 4 μl of gold fiducials (BSA Gold Tracer 10nm, AURION) were applied onto the grid, and after 30 s manually blotted with a piece of filter paper. Grids with applied cells were blotted (blotting force 0, blotting time 2 s, 100% humidity, wait time 15 s), plunge-frozen in liquid ethane using a Vitrobot Mark IV, and stored in liquid nitrogen."

Line 597 - should be "crYOLO"

A: Thank you. This has now been corrected (lines 699-700):

"A total of 24,991 phage capsids were automatically picked from micrographs of dataset 1 using the program crYOLO (Wagner et al., 2019)."

Line 1161 - should be "As pili retract"

A: Thank you. This has now been corrected (lines 1,330-1,331):

"(C) As pilus retracts, phage particle is brought to cell surface and binds it using receptor binding proteins."

Line 1164 - should be "form a channel"

A: Thank you, this has now been corrected (lines 1,332-1,333):

"(E) Tape measure proteins form a channel for genome translocation into host cell."

Line 1239 - should be "of the T5"

A: Thank you, this has now been corrected (lines 109-110):

"The sequence of the T5 the tape measure protein is marked: the zinc carboxypeptidase motif is shown in blue."

Comments on figures:

In a number of places, the points that are being made in the text are not entirely clear in the figure panels and some different/enlarged views or text indicators would be helpful e.g.

- line 82-83 (symmetry mismatch in Fig. 1)

A: We have now included symmetry indicators in Fig. 1A.

- line 97 (HK-97 fold in Fig. 2BD)

A: Thank you, this has now been included in panels 2BE (new panel order).

- line 110 (spine α -helix in Fig. 2B)

A: Thank you, this has now been included in 2E.

- line 111 (hooks interact with each other in Fig. 2B)

A: We have now included details of the interaction in Fig. 2C. Lines 1,216-1,218:

"(C) Cartoon representation of the interaction of major capsid proteins N-terminal hooks around twofold icosahedral axis of the capsid."

- line 181 (N-terminal whisker in Fig. 3E)

A: The whisker is now labeled in color in Fig. 3E.

Lines 1,251-1,255:

"Superimposition of cartoon representations of major capsid protein interacting with another capsomer (beige and N-terminus in dark blue) and capsid protein adjacent to portal dodecamer (purple and N-terminus in light blue) and one showing the bending of the N-terminal arm which enables incorporation of the portal complex into the capsid."

- line 232 (decoration domains... protrude tangentially in Fig. 4A)

A: We have now included a reference to Fig. 4B, where the domain is highlighted in color.

Lines 251-252:

"the decoration domains of JBD30 major tail protein have an immunoglobulin-like fold, and protrude tangentially from the major tail protein disc (Fig. 4AB)."

- line 287-289 - should this point to 5B? It is not apparent in A.

A: Thank you. This has now been corrected.

Lines 297-300:

"The linker runs from upper baseplate protein domain I to the baseplate interior, loops around adjacent baseplate hub domain II and returns to the outer baseplate surface, where it continues into upper baseplate protein domain II (Fig. 5B, S6EF)."

- line 170, 188 - label clip and stem domains in Fig. 3BC

A: Thank you. We have now included domain labels in Fig. 3B and C.

- line 189 - should this point to Fig. 3B?

A: Thank you. We have now included also a reference to 3B.

Lines 204-206:

"The C-terminus wedges into the groove between the clip domains of two adjacent portal proteins, continues along the stem domain α -helices, and ends at the beginning of the wing domain helix $\alpha 4$ (Fig. 3ABC)."

- line 226 - I take it that the hinge loop in Fig. 4B is the long b-hairpin - suggest using same nomenclature

A: Thank you, we have now unified the nomenclature to "hinge loop."

Lines 245-247:

"The major tail protein of JBD30 consists of a central β -sandwich (residues 8–41, 62–73, 184–256), peripheral α -helix (residues 77–83), decoration domain (residues 88–175), and a long hinge loop (residues 42–62) (Fig. 4B)."

- Fig. 2 - suggest using a different colour for the cysteines so they stand out

A: Thank you. We have now re-colored the cysteines to blue.

- Fig. 5F - what is the domain in grey? It seems misleading as T4P text label is also shown in

grey, but I think this should be beige. It would be helpful to label key domains in colour coded text.

A: Thank you, we have now included colored labels in Fig. 5F.

- Fig. 7C - the inset is more of a boxed area - the inset showing a putative channel needs to be enlarged

A: Thank you. We have now modified the figure as requested.

- Fig. S6 - I couldn't see that BppU was defined, and there are other abbreviations not mentioned in the legend

A: Thank you. We have now included an explanation of the abbreviations in the legend.

Lines 85-87:

"(E) Cartoon representation of upper baseplate protein (BppU) with domains coloured according to the sequence diagram at the bottom of the panel."

- Fig. S9 - what does "cell" refer to in the key?

A: Thank you. In this case "cell" was supposed to mean bacterium. We have now replaced "cell" in the plot description with "bacterium".

Dear Dr. Plevka,

Thank you for submitting a revised version of your manuscript. I sincerely apologise for the protracted assessment process due to delays in referee comment submission and the high number of submissions we receive at the moment.

Your study has now been seen by all original referees, who now find that most of their previous concerns have been addressed. Therefore, I would like to invite you to address the remaining referee points in the final revised version. I have discussed the remaining points by reviewer #2 with the other reviewers. Based on their input, please address these points as follows:

- 1) In response to point 1, please provide details on how the structure was obtained, including the particle number, reported resolution, and providing the maps and the corresponding FSC curves. Reviewer #1 suggests to also provide the structures with a 0.5 resolution threshold to avoid overfitting.
- 2) For point 2, reviewer #1 asks for more details to be provided in the methods about atomic model building. Since reviewer #2 is also unsure whether deposition of >5Å resolution structures in PDB is appropriate, I would be interested in discussing with you whether alternative databases, e.g., EMDB, would be more appropriate.

There are also a few editorial points that need to be addressed:

1. Please note the corresponding author in the manuscript text file.
2. Please check that the funding information is correct and identical both in the manuscript and our online system. Currently, Brno city municipality is missing in our system.
3. Please submit up to five keywords.
4. Please make sure that the order of the sections in the manuscript is as follows: complete author information, abstract, introduction, results, discussion, materials & methods, data availability section, acknowledgments, disclosure statement and competing interests, references, main figure legends, tables, expanded figure legends
5. CRedit has replaced the traditional author contributions section because it offers a systematic, machine-readable author contributions format that allows for more effective research assessment. Please remove the Authors Contributions from the manuscript and use the free text boxes beneath each contributing author's name in our online submission system to add specific details on the author's contribution. More information is available in our guide to authors.
6. Please rename "Inclusion & Ethics" section into "Disclosure and competing interests statement" (further info: <https://www.embopress.org/page/journal/14602075/authorguide#conflictsofinterest>).
7. Figure panels 1C-D, 4D, 7F-J, 8A-H are not mentioned in the manuscript text, please add the corresponding callouts.
8. In the Appendix, please add page numbers in the table of contents. Please update the nomenclature to Appendix Figure S1-S13 and Appendix Table S1-S3 throughout the manuscript and Appendix files.
9. Please define the scale bar for figures 1a, c.
10. In the Data Availability section, please add resolvable links to the datasets. More information about the format of this section can be found here: <https://www.embopress.org/page/journal/14602075/authorguide#dataavailability>.
11. In our standard image integrity check, we noted that the images appear to be reused between Figure 7 f,h,i,j and Figure S10 a,c,d,e. If this is intentional, please note the image reuse in the figure legend.

With best wishes,

Ieva

Ieva Gailite, PhD
Senior Scientific Editor
The EMBO Journal
Meyershofstrasse 1
D-69117 Heidelberg
Tel: +4962218891309
i.gailite@embojournal.org

We realize that it is difficult to revise to a specific deadline. In the interest of protecting the conceptual advance provided by the work, we recommend a revision within 3 months (6th Oct 2024). Please discuss the revision progress ahead of this time with the editor if you require more time to complete the revisions.

Referee #1:

Review

EMBOJ-2024-117341R P. Plevka and co-authors

The authors made a good job and have taken in account nearly all comments made by reviewers. The Ms is nearly ready for the publication. There are two comments:

Lines 166-167. Remove the sentence " The continuity of the strands is disrupted below fivefold vertexes of the capsid (Fig. S4A)." . The figure does not show any disruptions in the strands densities neither is the modified figure not in the previous one. This hypothetical suggestion is not supported by the data obtained and the significance of that is not clear.

Another comment is related to the figure Fig. S1. FSC curves of cryo-EM reconstructions. For the convenience of readers, possibly it would be useful to show the threshold 0.143 on all FSC panels.

My cordial congratulations to Dr P. Plevka and his team for the excellent study.

Referee #2:

The revised version of the MS "Carpe pili! Hunting strategy, structure, and replication of *P. aeruginosa* phage JBD30" is a much improved manuscript. The authors thoroughly addressed my concerns about functional aspects of their phage system. I am still uncertain about what is happening with the DAPI stain during phage infection - how the dye is incorporated into the new phage particles and what is happening with the host DNA in this case (why can't we see it?) - but this can be addressed in a separate project in the future.

I have to raise two technical concerns.

1. Table S2 (Data collection and structure quality indicators) lists an impossibly small number of particles that was used to calculate sub-4 Å resolution maps - e.g. the resolution of the C3 map of the baseplate is 3.6 Å with only 1,780 particles in it. This is impossible.

2. Fig. S2 shows resolution maps of various parts of the particle. The majority of the RBP and tail fiber maps have resolutions exceeding 5 Å, which does not allow for unambiguous atomic modeling. Certainly, these maps can be fitted using AlphaFold models, but this would not be an experimentally-based atomic model. It will still be a theoretical model of the complex, not much different from the original AlphaFold model. It's ok to use ribbon diagrams of such "experimentally-based" models in figures in the paper to support functional interpretation. However, depositing these theoretical models to the PDB as "experimental data" is completely different. This is a great disservice to current and future users of the PDB. My point: the PDB must contain atomic coordinates of domains in which cryoEM or X-ray maps for greater than 90% of residues show easily identifiable side chains. For this reason, I do not think that the atomic models of the RBP and tail fibers belong to the PDB.

Referee #3:

I am satisfied that the authors have made all of the changes that I suggested and this is a very nice paper. However, there are still some grammatical errors and I suggest the following improvements for consistency and readability:

59: phages requiring pili

347: Mechanism of JBD30 binding to the type IV pilus

381: the cell surface by pilus retraction

471: several cycles of pilus extension

488: showing that glycosylation of type IV pili blocks

501: result in reduced pilus production

1303: As the pilus retracts, the phage particle is brought to the cell surface and binds to it using receptor binding proteins.

1265: JBD30 baseplate bound to the type IV pilus

1266: density of the type IV pilus

1301: JBD30 virion attaches to the type IV pilus

1303: As the pilus retracts

The editor's and reviewer's comments are in blue italics, and our responses are in bold black font. Please note that the line numbers in this document refer to the manuscript file and supplementary material file with tracked changes, which were submitted as supplementary files for the revision process.

1) In response to point 1, please provide details on how the structure was obtained, including the particle number, reported resolution, and providing the maps and the corresponding FSC curves. Reviewer #1 suggests to also provide the structures with a 0.5 resolution threshold to avoid overfitting.

A: Thank you for considering this point in detail. Please see our response to reviewer #2 below. We have now included 0.143 and 0.5 resolution thresholds in all reconstruction FSC plots in Fig S1.

2) For point 2, reviewer #1 asks for more details to be provided in the methods about atomic model building. Since reviewer #2 is also unsure whether deposition of >5Å resolution structures in PDB is appropriate, I would be interested in discussing with you whether alternative databases, e.g., EMDB, would be more appropriate.

A: Please see our response to reviewer #2 below. We have already deposited all cryo-EM maps in EMDB (the codes are provided).

Referee #1:

EMBOJ-2024-117341R P. Plevka and co-authors

The authors made a good job and have taken in account nearly all comments made by reviewers. The Ms is nearly ready for the publication. There are two comments:
Lines 166-167. Remove the sentence "The continuity of the strands is disrupted below fivefold vertexes of the capsid (Fig. S4A)". The figure does not show any disruptions in the strands densities neither is the modified figure not in the previous one. This hypothetical suggestion is not supported by the data obtained and the significance of that is not clear.

A: We have now removed the sentence.

Another comment is related to the figure Fig. S1. FSC curves of cryo-EM reconstructions. For the convenience of readers, possibly it would be useful to show the threshold 0.143 on all FSC panels.

A: We have now included the 0.143 and 0.5 thresholds in all the panels in Appendix Figure S1.

My cordial congratulations to Dr P. Plevka and his team for the excellent study.

A: Thank you.

Referee #2:

The revised version of the MS "Carpe pili! Hunting strategy, structure, and replication of P. aeruginosa phage JBD30" is a much improved manuscript. The authors thoroughly addressed my concerns about functional aspects of their phage system. I am still uncertain about what is happening with the DAPI stain during phage infection - how the dye is incorporated into the new phage particles and what is happening with the host DNA in this case (why can't we see it?) - but this can be addressed in a separate project in the future.

I have to raise two technical concerns.

1. Table S2 (Data collection and structure quality indicators) lists an impossibly small number of particles that was used to calculate sub-4 Å resolution maps - e.g. the resolution of the C3 map of the baseplate is 3.6 Å with only 1,780 particles in it. This is impossible.

A: The initial number of particles used for the reconstruction was 8,376, after 2D and 3D classification we were indeed left with 1,780 particles that were used for the final reconstruction. Please note that the reconstruction employed threefold symmetry, effectively increasing the number of particles to 5,340. Also please note that the baseplate is relatively small (compared to for example a ribosome) and the amount of data required to fill the reciprocal space volume even at high resolution is thus limited. The indicators of the quality of the reconstruction are listed in Appendix Table S2. We deposited raw data in EMPIAR and anyone can independently re-calculate the reconstruction to verify/disprove our claims.

2. Fig. S2 shows resolution maps of various parts of the particle. The majority of the RBP and tail fiber maps have resolutions exceeding 5 Å, which does not allow for unambiguous atomic modeling. Certainly, these maps can be fitted using AlphaFold models, but this would not be an experimentally-based atomic model. It will still be a theoretical model of the complex, not much different from the original AlphaFold model. It's ok to use ribbon diagrams of such "experimentally-based" models in figures in the paper to support functional interpretation. However, depositing these theoretical models to the PDB as "experimental data" is completely different. This is a great disservice to current and future users of the PDB. My point: the PDB must contain atomic coordinates of domains in which cryoEM or X-ray maps for greater than 90% of residues show easily identifiable side chains. For this reason, I do not think that the atomic models of the RBP and tail fibers belong to the PDB.

A: Thank you for raising this concern. The overall resolutions of the reconstructions were actually 4.5 and 4.7Å. However, central parts of the domains were resolved to better than 4Å resolution, which enabled refinement of the initial models generated using AlphaFold2. We have now adjusted coloring in Appendix Figure S2 to display this clearly. Please, see the fit of the sidechains to the maps in Fig. R1 below.

Fig. R1. Fit of the refined PDB structures into cryo-EM reconstructions of (A) RBP and (B) tail fiber.

Referee #3:

I am satisfied that the authors have made all of the changes that I suggested and this is a very nice paper. However, there are still some grammatical errors and I suggest the following improvements for consistency and readability:

A: Thank you. We have now implemented all the grammatical corrections suggested by reviewer #3.

59: phages requiring pili

347: Mechanism of JBD30 binding to the type IV pilus

381: the cell surface by pilus retraction

471: several cycles of pilus extension

488: showing that glycosylation of type IV pili blocks

501: result in reduced pilus production

1303: As the pilus retracts, the phage particle is brought to the cell surface and binds to it using receptor binding proteins.

1265: JBD30 baseplate bound to the type IV pilus

1266: density of the type IV pilus

1301: JBD30 virion attaches to the type IV pilus

1303: As the pilus retracts

Dear Dr. Plevka,

Thank you for submitting the final revision of your manuscript. I have now looked into your response to the reviewers' comments, and I find it reasonable. I am now pleased to inform you that your manuscript has been accepted for publication - congratulations on a beautiful study!

Before we forward your manuscript to our publishers, I would like to propose some minor edits in the manuscript title, abstract and synopsis (please see below and the attached manuscript text file). I am afraid we found that the original title would have been more suitable to a News and Views or Review article and have rephrased in a more informational manner. I have also written a short blurb that will accompany the title of your manuscript in our online system. Please let me know if any corrections or adjustments are needed:

Title:

Structural snapshots elucidate the complete lifecycle of the *P. aeruginosa* phage JBD30

Blurb:

Cryo-electron microscopy structures of the siphophage Casadabanvirus JBD30 reveal its cell attachment, genome delivery, and virion assembly steps.

Synopsis:

To date, available structural insights into architecture and infection cycles of various bacteriophages, remain fragmented. Here, several cryo-electron microscopy structures reveal the complete lifecycle of the siphophage Casadabanvirus JBD30 during infection of its host *Pseudomonas aeruginosa*.

- JBD30 uses its baseplate to bind to the bacterial type-IV pilus, whose retraction brings JBD30 to the bacterial cell surface.
- The structure of the baseplate-pilus complex reveals a trimer of baseplate receptor-binding proteins that attach to the outer bacterial membrane.
- JBD30 major capsid proteins assemble into procapsids that undergo conformational changes to allow expansion upon filling with phage dsDNA.
- The tail of JBD30 bends easily due to flexible loops that mediate contacts between the successive discs of major tail proteins.

If you have any questions, please do not hesitate to contact the Editorial Office. Thank you for this contribution to The EMBO Journal and congratulations on a great paper!

With best wishes,

Ieva

Ieva Gailite, PhD
Senior Scientific Editor
The EMBO Journal
Meyerohofstrasse 1
D-69117 Heidelberg
Tel: +4962218891309
i.gailite@embojournal.org
